# C-Learning: Learning to Achieve Goals via Recursive Classification

**Benjamin Eysenbach**
CMU, Google Brain
beysenba@cs.cmu.edu

**Ruslan Salakhutdinov**
CMU

**Sergey Levine**
UC Berkeley, Google Brain

## Abstract

We study the problem of predicting and controlling the future state distribution of an autonomous agent. This problem, which can be viewed as a reframing of goal-conditioned reinforcement learning (RL), is centered around learning a conditional probability density function over future states. Instead of directly estimating this density function, we indirectly estimate this density function by training a classifier to predict whether an observation comes from the future. Via Bayes' rule, predictions from our classifier can be transformed into predictions over future states. Importantly, an off-policy variant of our algorithm allows us to predict the future state distribution of a new policy, without collecting new experience. This variant allows us to optimize functionals of a policy's future state distribution, such as the density of reaching a particular goal state. While conceptually similar to Q-learning, our work lays a principled foundation for goal-conditioned RL as density estimation, providing justification for goal-conditioned methods used in prior work. This foundation makes hypotheses about Q-learning, including the optimal goal-sampling ratio, which we confirm experimentally. Moreover, our proposed method is competitive with prior goal-conditioned RL methods.[1]

## 1 Introduction

In this paper, we aim to reframe the goal-conditioned reinforcement learning (RL) problem as one of *predicting* and *controlling* the future state of the world. This reframing is useful not only because it suggests a new algorithm for goal-conditioned RL, but also because it explains a commonly used heuristic in prior methods, and suggests how to automatically choose an important hyperparameter. The problem of predicting the future amounts to learning a probability density function over future states, agnostic of the time that a future state is reached. The future depends on the actions taken by the policy, so our predictions should depend on the agent's policy. While we could simply witness the future, and fit a density model to the observed states, we will be primarily interested in the following prediction question: Given experience collected from one policy, can we predict what states a *different* policy will visit? Once we can predict the future states of a different policy, we can *control* the future by choosing a policy that effects a desired future.

While conceptually similar to Q-learning, our perspective is different in that we make no reliance on reward functions. Instead, an agent can solve the prediction problem *before* being given a reward function, similar to models in model-based RL. Reward functions can require human supervision to construct and evaluate, so a fully autonomous agent can learn to solve this prediction problem before being provided any human supervision, and reuse its predictions to solve many different downstream tasks. Nonetheless, when a reward function is provided, the agent can estimate its expected reward under the predicted future state distribution. This perspective is different from prior approaches. For example, directly fitting a density model to future states only solves the prediction problem in the on-policy setting, precluding us from predicting where a different policy will go. Model-based approaches, which learn an explicit dynamics model, do allow us to predict the future state distribution of different policies, but require a reward function or distance metric to learn goal-reaching policies for controlling the future. Methods based on temporal difference (TD) learning (Sutton, 1988) have been used to predict the future state distribution (Dayan, 1993;

---

[1] Project website with videos and code: https://ben-eysenbach.github.io/c_learning/

Szepesvari et al., 2014; Barreto et al., 2017) and to learn goal-reaching policies (Kaelbling, 1993; Schaul et al., 2015). Section 3 will explain why these approaches do not learn a true Q function in continuous environments with sparse rewards, and it remains unclear what the learned Q function corresponds to. In contrast, our method will estimate a well defined classifier.

Since it is unclear how to use Q-learning to estimate such a density, we instead adopt a contrastive approach, learning a classifier to distinguish "future states" from random states, akin to Gutmann & Hyvärinen (2010). After learning this binary classifier, we apply Bayes' rule to obtain a probability density function for the future state distribution, thus solving our prediction problem. While this initial approach requires on-policy data, we then develop a bootstrapping variant for estimating the future state distribution for different policies. This bootstrapping procedure is the core of our goal-conditioned RL algorithm.

The main contribution of our paper is a reframing of goal-conditioned RL as estimating the probability density over future states. We derive a method for solving this problem, C-learning, which we use to construct a complete algorithm for goal-conditioned RL. Our reframing lends insight into goal-conditioned Q-learning, leading to a hypothesis for the optimal ratio for sampling goals, which we demonstrate empirically. Experiments demonstrate that C-learning more accurately estimates the density over future states, while remaining competitive with recent goal-conditioned RL methods across a suite of simulated robotic tasks.

## 2 RELATED WORK

Common goal-conditioned RL algorithms are based on behavior cloning (Ghosh et al., 2019; Ding et al., 2019; Gupta et al., 2019; Eysenbach et al., 2020; Lynch et al., 2020; Oh et al., 2018; Sun et al., 2019), model-based approaches (Nair et al., 2020; Ebert et al., 2018), Q-learning (Kaelbling, 1993; Schaul et al., 2015; Pong et al., 2018), and semi-parametric planning (Savinov et al., 2018; Eysenbach et al., 2019; Nasiriany et al., 2019; Chaplot et al., 2020). Most prior work on goal-conditioned RL relies on manually-specified reward functions or distance metric, limiting the applicability to high-dimensional tasks. Our method will be most similar to the Q-learning methods, which are applicable to off-policy data. These Q-learning methods often employ hindsight relabeling (Kaelbling, 1993; Andrychowicz et al., 2017), whereby experience is modified by changing the commanded goal. New goals are often taken to be a future state or a random state, with the precise ratio being a sensitive hyperparameter. We emphasize that our discussion of goal sampling concerns relabeling previously-collected experience, not on the orthogonal problem of sampling goals for exploration (Pong et al., 2018; Fang et al., 2019; Pitis et al., 2020).

Our work is closely related to prior methods that use TD-learning to predict the future state distribution, such as successor features (Dayan, 1993; Barreto et al., 2017; 2019; Szepesvari et al., 2014) and generalized value functions (Sutton & Tanner, 2005; Schaul et al., 2015; Schroecker & Isbell, 2020). Our approach bears a resemblance to these prior TD-learning methods, offering insight into why they work and how hyperparameters such as the goal-sampling ratio should be selected. Our approach differs in that it does not require a reward function or manually designed relabeling strategies, with the corresponding components being derived from first principles. While prior work on off-policy evaluation (Liu et al., 2018; Nachum et al., 2019) also aims to predict the future state distribution, our work differs is that we describe how to *control* the future state distribution, leading to goal-conditioned RL algorithm.

Our approach is similar to prior work on noise contrastive estimation (Gutmann & Hyvärinen, 2010), mutual-information based representation learning (Oord et al., 2018; Nachum et al., 2018), and variational inference methods (Bickel et al., 2007; Uehara et al., 2016; Dumoulin et al., 2016; Huszár, 2017; Sønderby et al., 2016). Like prior work on the probabilistic perspective on RL (Kappen, 2005; Todorov, 2008; Theodorou et al., 2010; Ziebart, 2010; Rawlik et al., 2013; Ortega & Braun, 2013; Levine, 2018), we treat control as a density estimation problem, but our main contribution is orthogonal: we propose a method for estimating the future state distribution, which can be used as a subroutine in both standard RL and these probabilistic RL methods.

## 3 PRELIMINARIES

We start by introducing notation and prior approaches to goal-conditioned RL. We define a controlled Markov process by an initial state distribution $p_1(\mathbf{s_1})$ and dynamics function $p(\mathbf{s_{t+1}} \mid \mathbf{s_t}, \mathbf{a_t})$.

We control this process by a Markovian policy $\pi_\theta(\mathbf{a_t} \mid \mathbf{s_t})$ with parameters $\theta$. We use $\pi_\theta(\mathbf{a_t} \mid \mathbf{s_t}, \mathbf{g})$ to denote a goal-oriented policy, which is additionally conditioned on a goal $\mathbf{g} \in \mathcal{S}$. We use $\mathbf{s_{t+}}$ to denote the random variable representing a future observation, defined by the following distribution:

**Definition 1.** *The future* $\gamma-$*discounted state density function is*

$$p_+^\pi(\mathbf{s_{t+}} \mid \mathbf{s_t}, \mathbf{a_t}) \triangleq (1 - \gamma) \sum_{\Delta=1}^{\infty} \gamma^\Delta p_\Delta^\pi(\mathbf{s_{t+\Delta}} = \mathbf{s_{t+}} \mid \mathbf{s_t}, \mathbf{a_t}),$$

*where $s_{t+\Delta}$ denotes the state exactly $\Delta$ in the future, and constant $(1 - \gamma)$ ensures that this density function integrates to 1.*

This density reflects the states that an agent would visit if we collected many infinite-length trajectories and weighted states in the near-term future more highly. Equivalently, $p(\mathbf{s_{t+}})$ can be seen as the distribution over terminal states we would obtain if we (hypothetically) terminated episodes at a random time step, sampled from a geometric distribution. We need not introduce a reward function to define the problems of predicting and controlling the future.

In discrete state spaces, we can convert the problem of estimating the future state distribution into a RL problem by defining a reward function $r_{\mathbf{s_{t+}}}(\mathbf{s_t}, \mathbf{a_t}) = \mathbb{1}(\mathbf{s_t} = \mathbf{s_{t+}})$, and terminating the episode when the agent arrives at the goal. The Q-function, which typically represents the expected discounted sum of future rewards, can then be interpreted as a (scaled) probability *mass* function:

$$Q^\pi(\mathbf{s_t}, \mathbf{a_t}, \mathbf{s_{t+}}) = \mathbb{E}_\pi \left[ \sum_t \gamma^t r_{\mathbf{s_{t+}}}(\mathbf{s_t}, \mathbf{a_t}) \right] = \sum_t \gamma^t \mathbb{P}_\pi(\mathbf{s_t} = \mathbf{s_{t+}}) = \frac{1}{1 - \gamma} p_+^\pi(\mathbf{s_{t+}} \mid \mathbf{s_t}, \mathbf{a_t}).$$

However, in continuous state spaces with some stochasticity in the policy or dynamics, the probability that any state *exactly* matches the goal state is zero.

**Remark 1.** *In a stochastic, continuous environment, for any policy $\pi$ the Q-function for the reward function $r_{\mathbf{s_{t+}}} = \mathbb{1}(\mathbf{s_t} = \mathbf{s_{t+}})$ is always zero:* $Q^\pi(\mathbf{s_t}, \mathbf{a_t}, \mathbf{s_{t+}}) = 0$.

This Q-function is not useful for predicting or controlling the future state distribution. Fundamentally, this problem arises because the relationship between the reward function, the Q function, and the future state distribution in prior work remains unclear. Prior work avoids this issue by manually defining reward functions (Andrychowicz et al., 2017) or distance metrics (Schaul et al., 2015; Pong et al., 2018; Zhao et al., 2019; Schroecker & Isbell, 2020). An alternative is to use hindsight relabeling, changing the commanded goal to be the goal actually reached. This form of hindsight relabeling does not require a reward function, and indeed learns Q-functions that are not zero (Lin et al., 2019). However, taken literally, Q-functions learned in this way must be incorrect: they do not reflect the expected discounted reward. An alternative hypothesis is that these Q-functions reflect probability *density* functions over future states. However, this also cannot be true:

**Remark 2.** *For any MDP with the sparse reward function $\mathbb{1}(\mathbf{s_t} = \mathbf{s_{t+}})$ where the episode terminates upon reaching the goal, Q-learning with hindsight relabeling acquires a Q-function in the range $Q^\pi(\mathbf{s_t}, \mathbf{a_t}, \mathbf{s_{t+}}) \in [0, 1]$, but the probability density function $p_+^\pi(\mathbf{s_{t+}} \mid \mathbf{s_t}, \mathbf{a_t})$ has a range $[0, \infty)$.*

For example, if the state space is $\mathcal{S} = [0, \frac{1}{2}]$, then there must exist some state $s_{t+}$ such that $Q^\pi(s_t, a_t, s_{t+1}) \leq 1 < p_+^\pi(\mathbf{s_{t+}} = s_{t+} \mid s_t, a_t)$. See Appendix H for two worked examples. Thus, Q-learning with hindsight relabeling also fails to learn the future state distribution. In fact, it is unclear what quantity Q-learning with hindsight relabeling optimizes. In the rest of this paper, we will define goal reaching in continuous state spaces in a way that is consistent and admits well-defined solutions (Sec. 4), and then present a practical algorithm for finding these solutions (Sec. 5).

## 4 FRAMING GOAL CONDITIONED RL AS DENSITY ESTIMATION

This section presents a novel framing of the goal-conditioned RL problem, which resolves the ambiguity discussed in the previous section. Our main idea is to view goal-conditioned RL as a problem of estimating the density $p_+^\pi(\mathbf{s_{t+}} \mid \mathbf{s_t}, \mathbf{a_t})$ over future states that a policy $\pi$ will visit, a problem that Q-learning does not solve (see Section 3). Section 5 will then explain how to use this estimated distribution as the core of a complete goal-conditioned RL algorithm.

**Definition 2.** *Given policy $\pi$, the **future state density estimation** problem is to estimate the $\gamma-$discounted state distribution of $\pi$:* $f_\theta^\pi(\mathbf{s_{t+}} \mid \mathbf{s_t}, \mathbf{a_t}) \approx p_+^\pi(\mathbf{s_{t+}} \mid \mathbf{s_t}, \mathbf{a_t})$.

The next section will show how to estimate $f_\theta^\pi$. Once we have found $f_\theta^\pi$, we can determine the probability that a future state belongs to a set $\mathcal{S}_{t+}$ by integrating over that set: $\mathbb{P}(\mathbf{s_{t+}} \in \mathcal{S}_{t+}) = \int f_\theta^\pi(\mathbf{s_{t+}} \mid \mathbf{s_t}, \mathbf{a_t}) \mathbb{1}(\mathbf{s_{t+}} \in \mathcal{S}_{t+}) d\mathbf{s_{t+}}$. Appendix A discusses a similar relationship with partially observed goals. There is a close connection between this integral and a goal-conditioned Q-function:

**Remark 3.** *For a goal $g$, define a reward function as an $\epsilon$-ball around the true goal: $r_g(\mathbf{s_t}, \mathbf{a_t}) = \mathbb{1}(\mathbf{s_{t+}} \in \mathcal{B}(g; \epsilon))$. Then the true Q-function is a scaled version of the probability density, integrated over the set $\mathcal{S}_{t+} = \mathcal{B}(g; \epsilon)$: $Q^\pi(\mathbf{s_t}, \mathbf{a_t}, g) = \mathbb{E}_\pi \left[ \sum_t \gamma^t r_g(s, a) \right] = (1 - \gamma) \mathbb{P}(\mathbf{s_{t+}} \in \mathcal{B}(g; \epsilon))$.*

## 5 C-LEARNING

We now derive an algorithm (C-learning) for solving the future state density estimation problem (Def. 2). First (Sec. 5.1), we assume that the policy is fixed, and present on-policy and off-policy solutions. Based on these ideas, Section 5.2 builds a complete goal-conditioned RL algorithm for learning an optimal goal-reaching policy. Our algorithm bears a resemblance to Q-learning, and our derivation makes two hypotheses about when and where Q-learning will work best (Sec. 5.3).

### 5.1 LEARNING THE CLASSIFIER

Rather than estimating the future state density directly, we will estimate it indirectly by learning a classifier. Not only is classification generally an easier problem than density estimation, but also it will allow us to develop an off-policy algorithm in the next section. We will call our approach *C-learning*. We start by deriving an on-policy Monte Carlo algorithm (*Monte Carlo C-learning*), and then modify it to obtain an off-policy, bootstrapping algorithm (*off-policy C-learning*). After learning this classifier, we can apply Bayes' rule to convert its binary predictions into future state density estimates. Given a distribution over state action pairs, $p(\mathbf{s_t}, \mathbf{a_t})$, we define the marginal future state distribution $p(\mathbf{s_{t+}}) = \int p_+^\pi(\mathbf{s_{t+}} \mid \mathbf{s_t}, \mathbf{a_t}) p(\mathbf{s_t}, \mathbf{a_t}) d\mathbf{s_t} d\mathbf{a_t}$. The classifier

---

**Algorithm 1 Monte Carlo C-learning**

**Input** trajectories $\{\tau_i\}$
Define $p(s, a) \leftarrow \text{Unif}(\{s, a\}_{(s,a) \sim \tau})$,
$\quad\quad p(s_{t+}) \leftarrow \text{Unif}(\{s_t\}_{s_t \sim \tau, t > 1})$
**while** not converged **do**
$\quad$ Sample $s_t, a_t \sim p(s, a), s_{t+}^{(0)} \sim p(s_{t+})$,
$\quad\quad\quad \Delta \sim \text{GEOM}(1 - \gamma)$.
$\quad$ Set goal $s_{t+}^{(1)} \leftarrow s_{t+\Delta}$
$\quad$ $\mathcal{F}(\theta) \leftarrow \log C_\theta^\pi(F = 1 \mid s_t, a_t, s_{t+}^{(1)})$
$\quad\quad\quad + \log C_\theta^\pi(F = 0 \mid s_t, a_t, s_{t+}^{(0)})$
$\quad$ $\theta \leftarrow \theta - \eta \nabla_\theta \mathcal{F}(\theta)$
**Return** classifier $C_\theta$

---

takes as input a state-action pair $(\mathbf{s_t}, \mathbf{a_t})$ together with another state $\mathbf{s_{t+}}$, and predicts whether $\mathbf{s_{t+}}$ was sampled from the future state density $p_+^\pi(\mathbf{s_{t+}} \mid \mathbf{s_t}, \mathbf{a_t})$ ($F = 1$) or the marginal state density $p(\mathbf{s_{t+}})$ ($F = 0$). The Bayes optimal classifier is

$$p(F = 1 \mid \mathbf{s_t}, \mathbf{a_t}, \mathbf{s_{t+}}) = \frac{p_+^\pi(\mathbf{s_{t+}} \mid \mathbf{s_t}, \mathbf{a_t})}{p_+^\pi(\mathbf{s_{t+}} \mid \mathbf{s_t}, \mathbf{a_t}) + p(\mathbf{s_{t+}})}. \tag{1}$$

Thus, using $C_\theta^\pi(F = 1 \mid \mathbf{s_t}, \mathbf{a_t}, \mathbf{s_{t+}})$ to denote our learned classifier, we can obtain an estimate $f_\theta^\pi(\mathbf{s_{t+}} \mid \mathbf{s_t}, \mathbf{a_t})$ for the future state density function using our classifier's predictions as follows:

$$f_\theta^\pi(\mathbf{s_{t+}} \mid \mathbf{s_t}, \mathbf{a_t}) = \frac{C_\theta^\pi(F = 1 \mid \mathbf{s_t}, \mathbf{a_t}, \mathbf{s_{t+}})}{C_\theta^\pi(F = 0 \mid \mathbf{s_t}, \mathbf{a_t}, \mathbf{s_{t+}})} p(\mathbf{s_{t+}}). \tag{2}$$

While our estimated density $f_\theta$ depends on the marginal density $p(\mathbf{s_{t+}})$, our goal-conditioned RL algorithm (Sec. 5.2) will note require estimating this marginal density. In particular, we will learn a policy that chooses the action $\mathbf{a_t}$ that maximizes this density, but the solution to this maximization problem does not depend on the marginal $p(\mathbf{s_{t+}})$.

We now present an on-policy approach for learning the classifier, which we call *Monte Carlo C-Learning*. After sampling a state-action pair $(\mathbf{s_t}, \mathbf{a_t}) \sim p(\mathbf{s_t}, \mathbf{a_t})$, we can either sample a future state $\mathbf{s_{t+}^{(1)}} \sim p_+^\pi(\mathbf{s_{t+}} \mid s_t, \mathbf{a_t})$ with a label $F = 1$, or sample $\mathbf{s_{t+}^{(0)}} \sim p(\mathbf{s_{t+}})$ with a label $F = 0$. We then train the classifier maximize log likelihood (i.e., the negative cross entropy loss):

$$\mathcal{F}(\theta) \triangleq \mathbb{E}_{\substack{\mathbf{s_t}, \mathbf{a_t} \sim p(\mathbf{s_t}, \mathbf{a_t}) \\ \mathbf{s_{t+}^{(1)}} \sim p_+^\pi(\mathbf{s_{t+}} | \mathbf{s_t}, \mathbf{a_t})}} [\log C_\theta^\pi(F = 1 \mid \mathbf{s_t}, \mathbf{a_t}, \mathbf{s_{t+}^{(1)}})] + \mathbb{E}_{\substack{\mathbf{s_t}, \mathbf{a_t} \sim p(\mathbf{s_t}, \mathbf{a_t}) \\ \mathbf{s_{t+}^{(0)}} \sim p(\mathbf{s_{t+}})}} [\log C_\theta^\pi(F = 0 \mid \mathbf{s_t}, \mathbf{a_t}, \mathbf{s_{t+}^{(0)}})]. \tag{3}$$

To sample future states, we note that the density $p_+^\pi(\mathbf{s_{t+}} \mid \mathbf{s_t}, \mathbf{a_t})$ is a weighted mixture of distributions $p(\mathbf{s_{t+\Delta}} \mid \mathbf{s_t}, \mathbf{a_t})$ indicating the future state exactly $\Delta$ steps in the future:

$$p_+^\pi(\mathbf{s_{t+}} \mid \mathbf{s_t}, \mathbf{a_t}) = \sum_{\Delta=0}^{\infty} p(s_{t+\Delta} \mid \mathbf{s_t}, \mathbf{a_t})p(\Delta) \qquad \text{where} \quad p(\Delta) = (1-\gamma)\gamma^\Delta = \text{GEOM}(\Delta; 1-\gamma),$$

where GEOM is the geometric distribution. Thus, we sample a future state $\mathbf{s_{t+}}$ via ancestral sampling: first sample $\Delta \sim \text{GEOM}(1 - \gamma)$ and then, looking at the trajectory containing $(\mathbf{s_t}, \mathbf{a_t})$, return the state that is $\Delta$ steps ahead of $(\mathbf{s_t}, \mathbf{a_t})$. We summarize Monte Carlo C-learning in Alg. 1.

While conceptually simple, this algorithm requires on-policy data, as the distribution $p_+^\pi(\mathbf{s_{t+}} \mid \mathbf{s_t}, \mathbf{a_t})$ depends on the current policy $\pi$ *and the commanded goal.* Even if we fixed the policy parameters, we cannot use experience collected when commanding one goal to learn a classifier for another goal. This limitation precludes an important benefit of goal-conditioned learning: the ability to readily share experience across tasks. To lift this limitation, the next section will develop a bootstrapped version of this algorithm that works with off-policy data.

We now extend the Monte Carlo algorithm introduced above to work in the off-policy setting, so that we can estimate the future state density for different policies. In the off-policy setting, we are given a dataset of transitions $(\mathbf{s_t}, \mathbf{a_t}, \mathbf{s_{t+1}})$ and a *new* policy $\pi$, which we will use to generate actions for the next time step, $\mathbf{a_{t+1}} \sim \pi(\mathbf{a_{t+1}} \mid \mathbf{s_{t+1}})$. The main challenge is sampling from $p_+^\pi(\mathbf{s_{t+}} \mid \mathbf{s_t}, \mathbf{a_t})$, which depends on the new policy $\pi$. We address this challenge in two steps. First, we note a recursive relationship between the future state density at the current time step and the next time step:

$$\underbrace{p_+^\pi(\mathbf{s_{t+}} = s_{t+} \mid \mathbf{s_t}, \mathbf{a_t})}_{\text{future state density at current time step}} = (1-\gamma)\underbrace{p(\mathbf{s_{t+1}} = s_{t+} \mid \mathbf{s_t}, \mathbf{a_t})}_{\text{environment dynamics}} + \gamma\mathbb{E}_{\substack{p(\mathbf{s_{t+1}}\mid\mathbf{s_t},\mathbf{a_t}),\\ \pi(\mathbf{a_{t+1}}\mid\mathbf{s_{t+1}})}}\left[\underbrace{p_+^\pi(\mathbf{s_{t+}} = s_{t+} \mid \mathbf{s_{t+1}}, \mathbf{a_{t+1}})}_{\text{future state density at next time step}}\right].$$

$$(4)$$

We can now rewrite our classification objective in Eq. 3 as

$$\mathcal{F}(\theta, \pi) = \mathbb{E}_{\substack{p(\mathbf{s_t},\mathbf{a_t}),\, p(\mathbf{s_{t+1}}\mid\mathbf{s_t},\mathbf{a_t}),\\ \pi(\mathbf{a_{t+1}}\mid\mathbf{s_{t+1}}),\, p_+^\pi(\mathbf{s_{t+}}\mid\mathbf{s_{t+1}},\mathbf{a_{t+1}})}} \left[(1-\gamma)\log C_\theta^\pi(F = 1 \mid \mathbf{s_t}, \mathbf{a_t}, \mathbf{s_{t+1}}) + \gamma\log C_\theta^\pi(F = 1 \mid \mathbf{s_t}, \mathbf{a_t}, \mathbf{s_{t+}})\right]$$
$$+ \mathbb{E}_{p(\mathbf{s_t},\mathbf{a_t}),\, p(\mathbf{s_{t+}})}\left[\log C_\theta^\pi(F = 0 \mid \mathbf{s_t}, \mathbf{a_t}, \mathbf{s_{t+}})\right]. \qquad (5)$$

This equation is different from the Monte Carlo objective (Eq. 3) because it depends on the new policy, but it still requires sampling from $p_+^\pi(\mathbf{s_{t+}} \mid \mathbf{s_{t+1}}, \mathbf{a_{t+1}})$, which also depends on the new policy. Our second step is to observe that we can estimate expectations that use $p_+^\pi(\mathbf{s_{t+}} \mid \mathbf{s_t}, \mathbf{a_t})$ by sampling from the marginal $\mathbf{s_{t+}} \sim p(\mathbf{s_{t+}})$ and then weighting those samples by an importance weight, which we can estimate using our learned classifier:

$$w(\mathbf{s_{t+1}}, \mathbf{a_{t+1}}, \mathbf{s_{t+}}) \triangleq \frac{p_+^\pi(\mathbf{s_{t+}} \mid \mathbf{s_{t+1}}, \mathbf{a_{t+1}})}{p(\mathbf{s_{t+}})} = \frac{C_\theta^\pi(F = 1 \mid \mathbf{s_{t+1}}, \mathbf{a_{t+1}}, \mathbf{s_{t+}})}{C_\theta^\pi(F = 0 \mid \mathbf{s_{t+1}}, \mathbf{a_{t+1}}, \mathbf{s_{t+}})}. \qquad (6)$$

The second equality is obtained by taking Eq. 2 and dividing both sides by $p(\mathbf{s_{t+}})$. In effect, these weights account for the effect of the new policy on the future state density. We can now rewrite our objective by substituting the identity in Eq. 6 for the $p(\mathbf{s_{t+}})$ term in the expectation in Eq. 5. The written objective is $\mathcal{F}(\theta, \pi) =$

$$\mathbb{E}_{\substack{p(\mathbf{s_t},\mathbf{a_t}),\, p(\mathbf{s_{t+1}}\mid\mathbf{s_t},\mathbf{a_t}),\\ p(\mathbf{s_{t+}}),\, \pi(\mathbf{a_{t+1}}\mid\mathbf{s_{t+1}})}}\big[(1-\gamma)\log C_\theta^\pi(F=1\mid\mathbf{s_t},\mathbf{a_t},\mathbf{s_{t+1}}) + \gamma\lfloor w(\mathbf{s_{t+1}},\mathbf{a_{t+1}},\mathbf{s_{t+}})\rfloor_{\text{sg}}\log C_\theta^\pi(F = 1 \mid \mathbf{s_t}, \mathbf{a_t}, \mathbf{s_{t+}})$$
$$+ \log C_\theta^\pi(F = 0 \mid \mathbf{s_t}, \mathbf{a_t}, \mathbf{s_{t+}})\big]. \qquad (7)$$

We use $\lfloor\cdot\rfloor_{\text{sg}}$ as a reminder that the gradient of an importance-weighted objective should not depend on the gradients of the importance weights. Intuitively, this loss says that next states should be labeled as positive examples, states sampled from the marginal should be labeled as negative examples, but *reweighted* states sampled from the marginal are positive examples.

**Algorithm summary.** Alg 2 reviews off policy C-learning, which takes as input a policy and a dataset of *transitions*. At each iteration, we sample a $(\mathbf{s_t}, \mathbf{a_t}, \mathbf{s_{t+1}})$ transition from the dataset, a potential future state $\mathbf{s_{t+}} \sim p(\mathbf{s_{t+}})$ and the next action $\mathbf{a_{t+1}} \sim \pi(\mathbf{a_{t+1}} \mid \mathbf{s_{t+1}}, \mathbf{s_{t+}})$. We compute the importance weight using the current estimate from the classifier, and then plug the importance weight into the loss from Eq. 3. We then update the classifier using the gradient of this objective.

**Algorithm 2 Off-Policy C-learning**

> **Input** transitions $\{s_t, a, s_{t+1}\}$, policy $\pi_\phi$
> **while** not converged **do**
>   Sample $(s_t, a_t, s_{t+1}) \sim p(s_t, a_t, s_{t+1})$,
>   $s_{t+} \sim p(s_{t+}), a_{t+1} \sim \pi_\phi(a_{t+1} \mid s_t, a_t)$
>   $w \leftarrow \texttt{stop\_grad}\left(\frac{C_\theta^\pi(F=1|s_{t+1}, a_{t+1}, s_{t+})}{C_\theta^\pi(F=0|s_{t+1}, a_{t+1}, s_{t+})}\right)$
>   $\mathcal{F}(\theta, \pi) \leftarrow (1-\gamma) \log C_\theta^\pi(F=1|s_t, a_t, s_{t+1})$
>     $+ \log C_\theta^\pi(F=0|s_t, a_t, s_{t+})$
>     $+ \gamma w \log C_\theta^\pi(F=1|s_t, a_t, s_{t+})$
>   $\theta \leftarrow \theta - \eta \nabla_\theta \mathcal{F}(\theta, \pi)$
> **Return** classifier $C_\theta^\pi$

**Algorithm 3 Goal-Conditioned C-learning**

> **Input** transitions $\{s_t, a, s_{t+1}\}$
> **while** not converged **do**
>   Sample $(s_t, a_t, s_{t+1}) \sim p(s_t, a_t, s_{t+1})$,
>   $s_{t+} \sim p(s_{t+}), a_{t+1} \sim \pi(a_{t+1} \mid s_t, a_t, s_{t+})$
>   $w \leftarrow \texttt{stop\_grad}\left(\frac{C_\theta^\pi(F=1|s_{t+1}, a_{t+1}, s_{t+})}{C_\theta^\pi(F=0|s_{t+1}, a_{t+1}, s_{t+})}\right)$
>   $\mathcal{F}(\theta, \pi) \leftarrow (1-\gamma) \log C_\theta^\pi(F=1|s_t, a_t, s_{t+1})$
>     $+ \log C_\theta^\pi(F=0|s_t, a_t, s_{t+})$
>     $+ \gamma w \log C_\theta^\pi(F=1|s_t, a_t, s_{t+})$
>   $\theta \leftarrow \theta - \eta \nabla_\theta \mathcal{F}(\theta, \pi)$
>   $\mathcal{G}(\phi) \leftarrow \mathbb{E}_{\pi_\phi(a_t|s_t, g=s_{t+})}[\log C_\theta^\pi(F=1|s_t, a_t, s_{t+})]$
>   $\phi \leftarrow \phi + \eta \nabla_\phi \mathcal{G}(\phi)$
> **Return** policy $\pi_\phi$

**C-learning Bellman Equations.** In Appendix D.1, we provide a convergence proof for off-policy C-learning in the tabular setting. Our proof hinges on the fact that the TD C-learning update rule has the same effect as applying the following (unknown) Bellman operator:

$$\frac{C_\theta^\pi(F=1 \mid \mathbf{s_t}, \mathbf{a_t}, \mathbf{s_{t+}})}{C_\theta^\pi(F=0 \mid \mathbf{s_t}, \mathbf{a_t}, \mathbf{s_{t+}})} = (1-\gamma)\frac{p(\mathbf{s_{t+1}} = \mathbf{s_{t+}} \mid \mathbf{s_t}, \mathbf{a_t})}{p(\mathbf{s_{t+}})} + \gamma \mathbb{E}_{\substack{p(\mathbf{s_{t+1}}|\mathbf{s_t}, \mathbf{a_t}), \\ \pi(\mathbf{a_{t+1}}|\mathbf{s_t})}} \left[ \frac{C_\theta^\pi(F=1 \mid \mathbf{s_{t+1}}, \mathbf{a_{t+1}}, \mathbf{s_{t+}})}{C_\theta^\pi(F=0 \mid \mathbf{s_{t+1}}, \mathbf{a_{t+1}}, \mathbf{s_{t+}})} \right]$$

This equation tells us that C-learning is equivalent to maximizing the reward function $r_{\mathbf{s_{t+}}}(\mathbf{s_t}, \mathbf{a_t}) = p(\mathbf{s_{t+1}} = \mathbf{s_{t+}} \mid \mathbf{s_t}, \mathbf{a_t})/p(\mathbf{s_{t+}})$, but does so without having to estimate either the dynamics $p(\mathbf{s_{t+1}} \mid \mathbf{s_t}, \mathbf{a_t})$ or the marginal distribution $p(\mathbf{s_t})$.

## 5.2 GOAL-CONDITIONED RL VIA C-LEARNING

We now build a complete algorithm for goal-conditioned RL based on C-learning. We will derive this algorithm in two steps. First, while Section 5.1 shows how to estimate the future state density of a single policy, for goal-conditioned RL we will want to estimate the future state density of a conditional policy, which may be conditioned on many goals. Second, we will discuss how to update a policy using the learned density.

To acquire a classifier for a goal-conditioned policy, we need to apply our objective function (Eq. 7) to all policies $\{\pi_\phi(a \mid s, g) \mid g \in \mathcal{S}\}$. We can do this efficiently by additionally conditioning the classifier and the policy on the *commanded* goal $g \in \mathcal{S}$. However, for learning a goal-reaching policy, we will only need to query the classifier on inputs where $\mathbf{s_{t+}} = g$. Thus, we only need to learn a classifier conditioned on inputs where $\mathbf{s_{t+}} = g$, resulting in the following objective:

$$\mathbb{E}_{\substack{p(\mathbf{s_t}, \mathbf{a_t}), \, p(\mathbf{s_{t+1}}|\mathbf{s_t}, \mathbf{a_t}), \\ p(\mathbf{s_{t+}}), \, \pi(\mathbf{a_{t+1}}|\mathbf{s_{t+1}}, \mathbf{g}=\mathbf{s_{t+}})}} [(1-\gamma) \log C_\theta^\pi(F=1 \mid \mathbf{s_t}, \mathbf{a_t}, \mathbf{s_{t+1}}) + \log C_\theta^\pi(F=0 \mid \mathbf{s_t}, \mathbf{a_t}, \mathbf{s_{t+}})$$
$$+ \gamma \lfloor w(\mathbf{s_{t+1}}, \mathbf{a_{t+1}}, \mathbf{s_{t+}}) \rfloor_{\text{sg}} \log C_\theta^\pi(F=1 \mid \mathbf{s_t}, \mathbf{a_t}, \mathbf{s_{t+}})]. \tag{8}$$

The **difference** between this objective and the one derived in Section 5.1 (Eq. 7) is that the next action is sampled from a goal-conditioned policy. The density function obtained from this classifier (Eq. 2) represents the future state density of $\mathbf{s_{t+}}$, given that the policy was commanded to reach goal $g = s_{t+}$: $f_\theta^\pi(\mathbf{s_{t+}} = s_{t+} \mid \mathbf{s_t}, \mathbf{a_t}) = p_+^\pi(\mathbf{s_{t+}} = s_{t+} \mid \mathbf{s_t}, \mathbf{a_t}, g = s_{t+})$.

Now that we can estimate the future state density of a goal-conditioned policy, our second step is to optimize the policy w.r.t. this learned density function. We need to define a reward function that says how good a particular future state density is for reaching a particular goal. While we can use any functional of future state density, a natural choice is the KL divergence between a Dirac density centered at the commanded goal and the future state density of the goal-conditioned policy:

$$-D_{\text{KL}}(\mathbb{1}(\mathbf{s_{t+}} = g) \parallel p_+^\pi(\mathbf{s_{t+}} \mid \mathbf{s_t}, \mathbf{a_t}, g)) = \log p_+^\pi(\mathbf{s_{t+}} = g \mid \mathbf{s_t}, \mathbf{a_t}, g).$$

Importantly, computing this KL only requires the future state density of the commanded goal. Since $p_+^\pi(\mathbf{s_{t+}} \mid \mathbf{s_t}, \mathbf{a_t}, g = \mathbf{s_{t+}})$ is a monotone increasing function of the classifier predictions (see Eq. 2), we can write the policy objective in terms of the classifier predictions:

$$\mathcal{G}(\phi) = \max_\phi \mathbb{E}_{\pi_\phi(\mathbf{a_t}|\mathbf{s_t}, g)}[\log C_\theta^\pi(F=1 \mid \mathbf{s_t}, \mathbf{a_t}, \mathbf{s_{t+}} = g)].$$

If we collect new experience during training, then the marginal distribution $p(\mathbf{s_{t+}})$ will change throughout training. While this makes the learning problem for the classifier non-stationary, the learning problem for the policy (whose solution is independent of $p(\mathbf{s_{t+}})$) remains stationary.

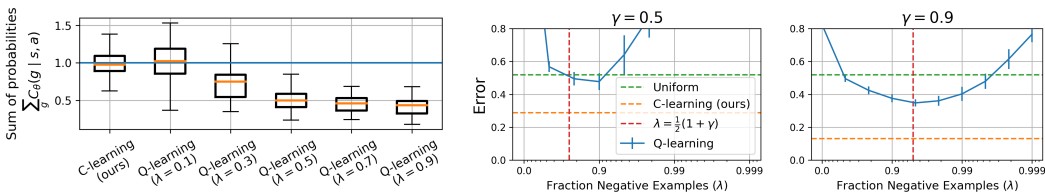

(a) Hypothesis 1: Underestimating the density    (b) Hypothesis 2: Optimal goal sampling ratio

Figure 1: **Testing Hypotheses about Q-learning**: *(Left)* As predicted, Q-values often sum to less than 1. *(Right)* The performance of Q-learning is sensitive to the relabeling ratio. Our analysis predicts that the optimal relabeling ratio is approximately $\lambda = \frac{1}{2}(1 + \gamma)$. C-learning (dashed orange) does not require tuning this ratio and outperforms Q-learning, even when the relabeling ratio for Q-learning is optimally chosen.

**Algorithm Summary:** We summarize our approach, which we call *goal-conditioned C-learning*, in Alg. 3. Given a dataset of transitions, we alternate between estimating the future state density of the goal-conditioned policy and updating the policy to maximize the probability density of reaching the commanded goal. This algorithm is simply to implement by taking a standard actor-critic RL algorithm and changing the loss function for the critic (a few lines of code). In the tabular setting, goal-conditioned C-learning converges to the optimal policy (proof in Appendix D.3).

## 5.3 Implications for Q-learning and Hindsight Relabeling

Off-policy C-learning (Alg. 2) bears a resemblance to Q-learning with hindsight relabeling, so we now compare these two algorithms to make hypotheses about Q-learning, which we will test in Section 6. We start by writing the objective for both methods using the cross-entropy loss, $\mathcal{CE}(\cdot, \cdot)$:

$$F_{\text{C-learning}}(\theta, \pi) = (1 - \gamma)\mathcal{CE}(C_\theta^\pi(F \mid \mathbf{s_t}, \mathbf{a_t}, \mathbf{s_{t+1}}), y = 1)$$

$$+ (1 + \gamma w)\mathcal{CE}\left(C_\theta^\pi(F \mid \mathbf{s_t}, \mathbf{a_t}, \mathbf{s_{t+}}), y = \frac{\gamma w}{\gamma w + 1} = \frac{\gamma C_\theta^{\pi'}}{\gamma C_\theta^{\pi'} + (1 - C_\theta^{\pi'})}\right), \quad (9)$$

$$F_{\text{Q-learning}}(\theta, \pi) = (1 - \lambda)\mathcal{CE}(Q_\theta^\pi(\mathbf{s_t}, \mathbf{a_t}, g = \mathbf{s_{t+1}}), y = 1)$$

$$+ \lambda \mathcal{CE}\left(Q_\theta^\pi(\mathbf{s_t}, \mathbf{a_t}, g = \mathbf{s_{t+}}), y = \gamma Q_\theta^\pi(\mathbf{s_{t+1}}, \mathbf{a_{t+1}}, \mathbf{s_{t+}})\right), \quad (10)$$

where $C_\theta' = C_\theta^\pi(F = 1 \mid \mathbf{s_{t+1}}, \mathbf{a_{t+1}}, \mathbf{s_{t+}})$ is the classifier prediction at the next state and where $\lambda \in [0, 1]$ denotes the *relabeling ratio* used in Q-learning, corresponding to the fraction of goals sampled from $p(\mathbf{s_{t+}})$. There are two differences between these equations, which lead us to make two hypotheses about the performance of Q-learning, which we will test in Section 6. The first difference is how the *predicted* targets are scaled for random goals, with Q-learning scaling the prediction by $\gamma$ while C-learning scales the prediction by $\gamma/(\gamma C_\theta' + (1 - C_\theta'))$. Since Q-learning uses a smaller scale, we make the following hypothesis:

**Hypothesis 1.** *Q-learning will predict **smaller** future state densities and therefore **underestimate** the true future state density function.*

This hypothesis is interesting because it predicts that prior methods based on Q-learning will not learn a proper density function, and therefore fail to solve the future state density estimation problem. The second difference between C-learning and Q-learning is that Q-learning contains a tunable parameter $\lambda$, which controls the ratio with which next-states and random states are used as goals. This ratio is equivalent to a weight on the two loss terms, and our experiments will show that Q-learning with hindsight relabeling is sensitive to this parameter. In contrast, C-learning does not require specification of this hyperparameter. Matching the coefficients in the Q-learning loss (Eq. 10) with those in our loss (Eq. 9) (i.e., $[1 - \lambda, \lambda] \propto [1 - \gamma, 1 + \gamma w]$), we make the following hypothesis:

**Hypothesis 2.** *Q-learning with hindsight relabeling will most accurately solve the future state density estimation problem (Def. 2) when random future states are sampled with probability $\lambda = \frac{1+\gamma}{2}$.*

Prior work has found that this goal sampling ratio is a sensitive hyperparameter (Andrychowicz et al., 2017; Pong et al., 2018; Zhao et al., 2019); this hypothesis is useful because it offers an automatic way to choose the hyperparameter. The next section will experimentally test these hypotheses.

## 6 Experiments

We aim our experiments at answering the following questions:

1. Do Q-learning and C-learning accurately estimate the future state density (Problem 2)?

2. (Hypothesis 1) Does Q-learning underestimate the future state density function (§ 5.3)?
3. (Hypothesis 2) Is the predicted relabeling ratio $\lambda = (1 + \gamma)/2$ optimal for Q-learning (§ 5.3)?
4. How does C-learning compare with prior goal-conditioned RL methods on benchmark tasks?

**Do Q-learning and C-learning accurately predict the future?** Our first experiment studies how well Q-learning and C-learning solve the future state density estimation problem (Def. 2). We use a continuous version of a gridworld for this task and measure how close the predicted future state density is to the true future state density using a KL divergence. Since this environment is continuous and stochastic, Q-learning without hindsight relabelling learns $Q = 0$ on this environment. In the on-policy setting, MC C-learning and TD C-learning perform similarly, while the prediction error for Q-learning (with hindsight relabeling) is more than three times worse. In the off-policy setting, TD C-learning is more accurate than Q-learning (with hindsight relabeling), achieving a KL divergence that is 14% lower than that of Q-learning. As expected, TD C-learning performs better than MC C-learning in the off-policy setting. These experiments demonstrate that C-learning yields a more accurate solution to the future state density estimation problem, as compared with Q-learning. See Appendix G.1 for full experimental details and results.

Our next experiment studies the ability of C-learning to predict the future in higher-dimensional continuous control tasks. We collected a dataset of experience from agents pre-trained to solve three locomotion tasks from OpenAI Gym. We applied C-learning to each dataset, and used the resulting classifier to predict the expected future state. As a baseline, we trained a 1-step dynamics model on this same dataset and unrolled this model autoregressively to obtain a prediction for the expected future state. Varying the discount factor,

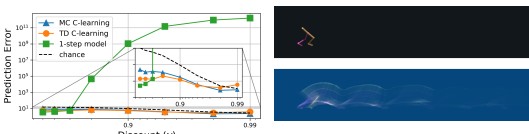

(a) Walker2d-v2       (b) Predicted future states

Figure 2: **Predicting the Future**: C-learning makes accurate predictions of the expected future state across a range of tasks and discount values. In contrast, learning a 1-step dynamics model and unrolling that model results in high error for large discount values.

we compared each method on `Walker2d-v2` in Fig. 2 and the other tasks in Appendix Fig. 7. The 1-step dynamics model is accurate over short horizons but performance degrades for larger values of $\gamma$, likely due to prediction errors accumulating over time. In contrast, the predictions obtained by MC C-learning and TD C-learning remain accurate for large values of $\gamma$. Appendix G.2 contains for experimental details; Appendix I and the project website contain more visualizations.

**Testing our hypotheses about Q-learning**: We now test two hypotheses made in Section 5.3. The first hypothesis is that Q-learning will underestimate the future state density function. To test this hypothesis, we compute the sum over the predicted future state density function, $\int_{s_{t+}} p_+^\pi(\mathbf{s_{t+}} = s_{t+} \mid \mathbf{s_t}, \mathbf{a_t})$, which in theory should equal one. We compared the predictions from MC C-learning and Q-learning using on-policy data (details in Appendix G.1). As shown in Fig. 1a, the predictions from C-learning summed to 1, but the predictions from Q-learning consistently summed to less than one, especially for large values of $\lambda$. However, our next experiment shows that Q-learning works best when using large values of $\lambda$, suggesting that successful hyperparameters for Q-learning are ones for which Q-learning does not learn a proper density function.

Our second hypothesis is that Q-learning will perform best when the relabeling ratio is chosen to be $\lambda = (1 + \gamma)/2$. Fig. 1b shows the results from this experiment. The performance of Q-learning is highly sensitive to the relabeling ratio: values of $\lambda$ that are too large or too small result in Q-learning performing poorly, worse than simply predicting a uniform distribution. Second, not only does the optimal choice of $\lambda$ increase with $\gamma$, but our theoretical hypothesis of $\lambda = (1 - \gamma)/2$ almost exactly predicts the optimal value of $\lambda$. Our third observation is that C-learning, which uses a 50-50 sampling ratio, consistently does better than Q-learning, even for the best choice of $\lambda$. These experiments support our hypothesis for the choice of relabeling ratio while reaffirming that our principled approach to future state density estimation obtains a more accurate solution.

**Goal-Conditioned RL for continuous control tasks**: Our last set of experiments apply goal-conditioned C-learning (Alg. 3) to benchmark continuous control tasks from prior work, shown in Fig. 3. These tasks range in difficulty from the 6-dimensional Sawyer Reach task to the 45-dimensional Pen task (see Appendix G). The aim of these experiments is to show that C-learning is competitive with prior goal-conditioned RL methods, without requiring careful tuning of the goal sampling ratio. We compare C-learning with a number of prior methods based on Q-learning, which

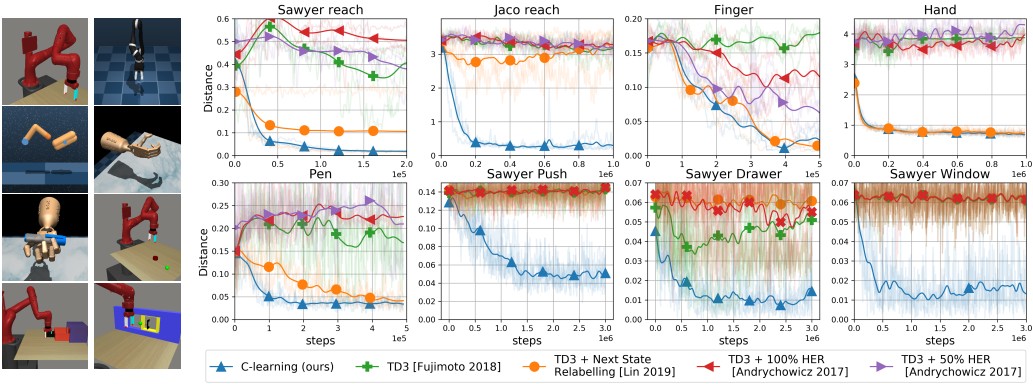

Figure 3: **Goal-conditioned RL**: C-learning is competitive with prior goal-conditioned RL methods across a suite of benchmark tasks, without requiring careful tuning of the relabeling distribution.

differ in how goals are sampled during training: TD3 (Fujimoto et al., 2018) does no relabeling, Lin et al. (2019) uses 50% next state goals and 50% random goals, and HER (Andrychowicz et al., 2017) uses final state relabeling (we compare against both 100% and 50% relabeling). None of these methods require a reward function or distance function for training; for evaluation, we use the L2 metric between the commanded goal and the terminal state (the average distance to goal and minimum distance to goal show the same trends). As shown in Fig. 3, C-learning is competitive with the best of these baselines across all tasks, and substantially better than all baselines on the Sawyer manipulation tasks. These manipulation tasks are more complex than the others because they require indirect manipulation of objects in the environment. Visualizing the learned policies, we observe that C-learning has discovered regrasping and fine-grained adjustment behaviors, behaviors that typically require complex reward functions to learn (Popov et al., 2017).[2] On the Sawyer Push and Sawyer Drawer tasks, we found that a hybrid of TD C-learning and MC C-learning performed better than standard C-learning. This variant, which is analogous to an "n-step" version of C-learning, is simple to implement and is described in Appendix E. In summary, C-learning performs as well as prior methods on simpler tasks and better on complex tasks, does not depend on a sensitive hyperparameter (the goal sampling ratio), and maximizes a well-defined objective function.

**Predicting the goal sampling ratio for goal conditioned RL**: While C-learning prescribes a precise method for sampling goals, prior hindsight relabeling methods are sensitive to these parameters. To visualize this, we varied the goal sampling ratio used by Lin et al. (2019) on the `maze2d-umaze-v0` task from Fu et al. (2020). As shown in Fig. 4, properly choosing this ratio can result in a 50% decrease in final distance. Additionally, our hypothesis that the optimal goal sampling ratio is $\lambda = (1 - \gamma)/2$ accurately predicts the best value for this ratio.

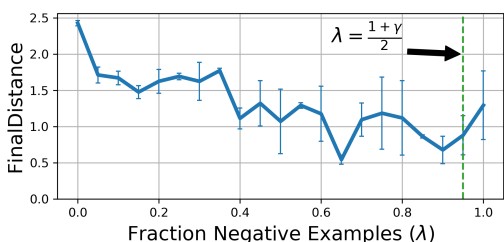

Figure 4: Q-learning is sensitive to the relabeling ratio. Our analysis predicts the optimal relabeling ratio.

## 7 CONCLUSION

A goal-oriented agent should be able to predict and control the future state of its environment. In this paper, we used this idea to reformulate the standard goal-conditioned RL problem as one of estimating and optimizing the future state density function. We showed that Q-learning does not directly solve this problem in (stochastic) environments with continuous states, and hindsight relabeling produces, at best, a mediocre solution for an unclear objective function. In contrast, C-learning yields more accurate solutions. Moreover, our analysis makes two hypotheses about when and where hindsight relabeling will most effectively solve this problem, both of which are validated in our experiments. Our experiments also demonstrate that C-learning scales to high-dimensional continuous controls tasks, where performance is competitive with state-of-the-art goal conditioned RL methods while offering an automatic and principled mechanism for hindsight relabeling.

---

[2]See the project website for videos: `https://ben-eysenbach.github.io/c_learning`

ACKNOWLEDGEMENTS

We thank Dibya Ghosh and Vitchyr Pong for discussions about this work, and thank Vincent Vanhouke, Ofir Nachum, and anonymous reviewers for providing feedback on early versions of this work. This work is supported by the Fannie and John Hertz Foundation, and the National Science Foundation (DGE-1745016, IIS1763562), and the US Army (W911NF1920104).

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

# A PARTIALLY-OBSERVED GOALS

In many realistic scenarios we have uncertainty over the true goal, with many goals having some probability of being the user's true goal. These scenarios might arise because of sensor noise or because the user wants the agent to focus on a subset of the goal observation (e.g., a robot's center of mass).

**Noisy Goals.** In many prior works, this setting is approached by assuming that a noisy measurement of the goal $z \sim p(z \mid \mathbf{s_{t+}} = g)$ is observed, and conditioning the Q-function on this measurement. For example, the measurement $z$ might be the VAE latent of the image $g$ (Pong et al., 2019; Gregor et al., 2018; Shelhamer et al., 2016). In this setting, we will instead aim to estimate the future discounted *measurement* distribution,

$$p(\mathbf{z_{t+}} \mid \mathbf{s_t}, \mathbf{a_t}) = (1 - \gamma) \sum_{\Delta=0}^{\infty} \gamma^{\Delta} \mathbb{E}_{\mathbf{s_{t+\Delta}} \sim p(\mathbf{s_{t+\Delta}} \mid \mathbf{s_t}, \mathbf{a_t})} \left[ p(\mathbf{z_{t+\Delta}} \mid \mathbf{s_{t+\Delta}}) \right].$$

Whereas the goal-conditioned setting viewed $f_\theta^\pi(\mathbf{s_{t+}} \mid \mathbf{s_t}, \mathbf{a_t})$ as defining a measure over *goals*, here we view $f_\theta^\pi(\mathbf{z_{t+}} \mid \mathbf{s_t}, \mathbf{a_t})$ as an implicitly-defined distribution over *measurements*.

**Partial Goals.** In some settings, the user may want the agent to pay attention to part of the goal, ignoring certain coordinates or attributes. Applying C-learning to this setting is easy. Let $\mathtt{crop}(\mathbf{s_t})$ be a user-provided function that extracts relevant coordinates or aspects of the goal state. Then, the user can simply parametrize the C-learning classifier as $C_\theta^\pi(F \mid \mathbf{s_t}, \mathbf{a_t}, g = \mathtt{crop}(\mathbf{s_{t+}}))$.

# B ANALYTIC FUTURE STATE DISTRIBUTION

In this section we describe how to analytically compute the discounted future state distribution for the gridworld examples. We started by creating two matrices:

$$T \in \mathbb{R}^{25 \times 25} : \quad T[s, s'] = \sum_a \mathbb{1}(f(s, a) = s') \pi(a \mid s)$$

$$T_0 \in \mathbb{R}^{25 \times 4 \times 25} : \quad T[s, a, s'] = \mathbb{1}(f(s, a) = s'),$$

where $f(s, a)$ denotes the deterministic transition function. The future discounted state distribution is then given by:

$$\begin{aligned} P &= (1 - \gamma) \left[ T_0 + \gamma T_0 T + \gamma^2 T_0 T^2 + \gamma^3 T_0 T^3 + \cdots \right] \\ &= (1 - \gamma) T_0 \left[ I + \gamma T + \gamma^2 T^2 + \gamma^3 T^3 + \cdots \right] \\ &= (1 - \gamma) T_0 \left( I - \gamma T \right)^{-1} \end{aligned}$$

The tensor-matrix product $T_0 T$ is equivalent to $\mathtt{einsum}$('ijk,kh $\to$ ijh', $T_0, T$). We use the forward KL divergence for estimating the error in our estimate, $D_{\mathrm{KL}}(P \parallel Q)$, where $Q$ is the tensor of predictions:

$$Q \in \mathbb{R}^{25 \times 4 \times 25} : \quad Q[s, a, g] = q(g \mid s, a).$$

# C ASSIGNMENT EQUATIONS FOR THE MSE LOSS

In Section 5.3, we derived the assignment equations for C-learning under the cross entropy loss. In this section we show that using the mean squared error (MSE) loss results in the same update equations. Equivalently, this can be viewed as using a Gaussian model for Q values instead of a logistic model. This result suggests that the difference in next-state scaling between C-learning and Q-learning is not just a quirk of the loss function.

To start, we write the loss for C-learning using the MSE and then completing the square.

$$
\begin{aligned}
L(\theta, \pi) &= (1-\gamma)(C_\theta^\pi(F=1 \mid \mathbf{s_t}, \mathbf{a_t}, \mathbf{s_{t+1}}) - 1)^2 + \gamma w(C_\theta^\pi(F=1 \mid \mathbf{s_t}, \mathbf{a_t}, \mathbf{s_{t+}}) - 1)^2 \\
&\quad + (C_\theta^\pi(F=1 \mid \mathbf{s_t}, \mathbf{a_t}, \mathbf{s_{t+1}}) - 0)^2 \\
&= (1-\gamma)\left(C_\theta^\pi(F=1 \mid \mathbf{s_t}, \mathbf{a_t}, \mathbf{s_{t+1}}) - 1\right)^2 + \gamma w\left(C_\theta^\pi(F=1 \mid \mathbf{s_t}, \mathbf{a_t}, \mathbf{s_{t+}}) - \frac{\gamma w}{\gamma w + 1}\right)^2 \\
&\quad + \underbrace{\gamma w - \left(\frac{\gamma w}{\gamma w + 1}\right)^2}_{\text{constant w.r.t. } C_\theta}.
\end{aligned}
$$

The optimal values for $C_\theta$ for both cases of goal are the same as for the cross entropy loss:

$$
C_\theta^\pi(F=1 \mid \mathbf{s_t}, \mathbf{a_t}, \mathbf{s_{t+}}) \leftarrow \begin{cases} 1 & \text{if } \mathbf{s_{t+1}} = \mathbf{s_{t+}} \\ \frac{\gamma w}{\gamma w + 1} & \text{otherwise} \end{cases}.
$$

Additionally, observe that the weights on the two loss terms are the same as for the cross entropy loss: the next-state goal loss is scaled by $(1-\gamma)$ while the random goal loss is scaled by $\gamma w$.

## D  A BELLMAN EQUATION FOR C-LEARNING AND CONVERGENCE GUARANTEES

The aim of this section is to show that off-policy C-learning converges, and that the fixed point corresponds to the Bayes-optimal classifier. This result guarantees that C-learning will accurate *evaluate* the future state density of a given policy. We then provide a policy improvement theorem, which guarantees that goal-conditioned C-learning converges to the optimal goal-conditioned policy.

### D.1  BELLMAN EQUATIONS FOR C-LEARNING

We start by introducing a new Bellman equation for C-learning, which will be satisfied by the Bayes optimal classifier. While actually evaluating this Bellman equation requires privileged knowledge of the transition dynamics and the marginal state density, if we knew these quantities we could turn this Bellman equation into a convergent value iteration procedure. In the next section, we will show that the updates of off-policy C-learning are equivalent to this value iteration procedure, but do not require knowledge of the transition dynamics or marginal state density. This equivalence allows us to conclude that C-learning converges to the Bayes-optimal classifier.

Our Bellman equation says that the future state density function $f_\theta$ induced by a classifier $C_\theta$ should satisfy the recursive relationship noted in Eq. 4.

**Lemma 1** (C-learning Bellman Equation). *Let policy $\pi(\mathbf{a_t} \mid \mathbf{s_t})$, dynamics function $p(\mathbf{s_{t+1}} \mid \mathbf{s_t}, \mathbf{a_t})$, and marginal distribution $p(\mathbf{s_{t+}})$ be given. If a classifier $C_\theta$ is the Bayes-optimal classifier, then it satisfies the follow identity for all states $\mathbf{s_t}$, actions $\mathbf{a_t}$, and potential future states $\mathbf{s_{t+}}$:*

$$
\frac{C_\theta^\pi(F=1 \mid \mathbf{s_t}, \mathbf{a_t}, \mathbf{s_{t+}})}{C_\theta^\pi(F=0 \mid \mathbf{s_t}, \mathbf{a_t}, \mathbf{s_{t+}})} = (1-\gamma)\frac{p(\mathbf{s_{t+1}} = \mathbf{s_{t+}} \mid \mathbf{s_t}, \mathbf{a_t})}{p(\mathbf{s_{t+}})} + \gamma \mathbb{E}_{\substack{p(\mathbf{s_{t+1}} \mid \mathbf{s_t}, \mathbf{a_t}), \\ \pi(\mathbf{a_{t+1}} \mid \mathbf{s_t})}} \left[ \frac{C_\theta^\pi(F=1 \mid \mathbf{s_{t+1}}, \mathbf{a_{t+1}}, \mathbf{s_{t+}})}{C_\theta^\pi(F=0 \mid \mathbf{s_{t+1}}, \mathbf{a_{t+1}}, \mathbf{s_{t+}})} \right]
\tag{11}
$$

*Proof.* If $C_\theta$ is the Bayes-optimal classifier, then $f_\theta^\pi(\mathbf{s_{t+}} \mid \mathbf{s_t}, \mathbf{a_t}) = p_+^\pi(\mathbf{s_{t+}} \mid \mathbf{s_t}, \mathbf{a_t})$. Substituting the definition of $f_\theta$ (Eq. 2) into Eq. 4, we obtain a new Bellman equation: $\qquad \square$

This Bellman equation is similar to the standard Bellman equation with a goal-conditioned reward function $r_{\mathbf{s_{t+}}}(\mathbf{s_t}, \mathbf{a_t}) = p(\mathbf{s_{t+1}} = \mathbf{s_{t+}} \mid \mathbf{s_t}, \mathbf{a_t})/p(\mathbf{s_{t+}})$, where Q functions represent the ratio $f_\theta^\pi(\mathbf{s_{t+}} \mid \mathbf{s_t}, \mathbf{a_t}) = p_+^\pi(\mathbf{s_{t+}} \mid \mathbf{s_t}, \mathbf{a_t})$. However, actually computing this reward function to evaluate this Bellman equation requires knowledge of the densities $p(\mathbf{s_{t+1}} \mid \mathbf{s_t}, \mathbf{a_t})$ and $p(\mathbf{s_{t+}})$, both of which we assume are unknown to our agent.[3] Nonetheless, if we had this privileged information,

---

[3]Interestingly, we can efficiently estimate this reward function by learning a *next-state* classifier, $q_\theta(F \mid \mathbf{s_t}, \mathbf{a_t}, \mathbf{s_{t+1}})$, which distinguishes $\mathbf{s_{t+1}} \sim p(\mathbf{s_{t+1}} \mid \mathbf{s_t}, \mathbf{a_t})$ from $\mathbf{s_{t+1}} \sim p(\mathbf{s_{t+1}}) = p(\mathbf{s_{t+}})$. This classifier

we could readily turn this Bellman equation into the following assignment equation:

$$\frac{C_\theta^\pi(F=1\mid \mathbf{s_t},\mathbf{a_t},\mathbf{s_{t+}})}{C_\theta^\pi(F=0\mid \mathbf{s_t},\mathbf{a_t},\mathbf{s_{t+}})} \leftarrow (1-\gamma)\frac{p(\mathbf{s_{t+1}}=\mathbf{s_{t+}}\mid \mathbf{s_t},\mathbf{a_t})}{p(\mathbf{s_{t+}})} + \gamma\mathbb{E}_{\substack{p(\mathbf{s_{t+1}}\mid \mathbf{s_t},\mathbf{a_t}),\\ \pi(\mathbf{a_{t+1}}\mid \mathbf{s_t})}}\left[\frac{C_\theta^\pi(F=1\mid \mathbf{s_{t+1}},\mathbf{a_{t+1}},\mathbf{s_{t+}})}{C_\theta^\pi(F=0\mid \mathbf{s_{t+1}},\mathbf{a_{t+1}},\mathbf{s_{t+}})}\right]$$

(12)

**Lemma 2.** *If we use a tabular representation for the* ratio $\frac{C_\theta^\pi(F=1\mid \mathbf{s_t},\mathbf{a_t},\mathbf{s_{t+}})}{C_\theta^\pi(F=0\mid \mathbf{s_t},\mathbf{a_t},\mathbf{s_{t+}})}$, *then iterating the assignment equation (Eq. 12) converges to the optimal classifier.*

*Proof.* Eq. 12 can be viewed as doing value iteration with a goal-conditioned Q function parametrized as $Q(\mathbf{s_t},\mathbf{a_t},\mathbf{s_{t+}}) = \frac{C_\theta^\pi(F=1\mid \mathbf{s_t},\mathbf{a_t},\mathbf{s_{t+}})}{C_\theta^\pi(F=0\mid \mathbf{s_t},\mathbf{a_t},\mathbf{s_{t+}})}$ and a goal-conditioned reward function $r_{\mathbf{s_{t+}}}(\mathbf{s_t},\mathbf{a_t}) = \frac{p(\mathbf{s_{t+1}}=\mathbf{s_{t+}}\mid \mathbf{s_t},\mathbf{a_t})}{p(\mathbf{s_{t+}})}$. We can then employ standard convergence proofs for Q-learning to guarantee convergence (Jaakkola et al., 1994, Theorem 1). $\square$

### D.2 OFF-POLICY C-LEARNING CONVERGES

In this section we show that off-policy C-learning converges to the Bayes-optimal classifier, and thus recovers the true future state density function. The main idea is to show that the updates for off-policy C-learning have the same effect as the assignment equation above (Eq. 12), without relying on knowledge of the dynamics function or marginal density function.

**Lemma 3.** *Off-policy C-learning results in the same updates to the classifier as the assignment equations for the C-learning Bellman equation (Eq. 12)*

*Proof.* We start by viewing the off-policy C-learning loss (Eq. 9) as a *probabilistic* assignment equation. A given triplet $(\mathbf{s_t},\mathbf{a_t},\mathbf{s_{t+}})$ can appear in Eq. 9 in two ways:

1. We sample a "positive" $\mathbf{s_{t+}} = \mathbf{s_{t+1}}$, which happens with probability $(1-\gamma)p(\mathbf{s_{t+1}} = \mathbf{s_{t+}}\mid \mathbf{s_t},\mathbf{a_t})$, and results in the label $y=1$.

2. We sample a "negative" $\mathbf{s_{t+}}$, which happens with probability $(1+\gamma w)p(\mathbf{s_{t+}})$ and results in the label $y = \frac{\gamma w}{\gamma w+1}$.

Thus, conditioned on the given triplet containing $\mathbf{s_{t+}}$, the expected target value $y$ is

$$\mathbb{E}[y\mid \mathbf{s_t},\mathbf{a_t},\mathbf{s_{t+}}] = \frac{(1-\gamma)p(\mathbf{s_{t+1}}=\mathbf{s_{t+}}\mid \mathbf{s_t},\mathbf{a_t})\cdot 1 + \mathbb{E}\left[\cancel{(1+\gamma w)}p(\mathbf{s_{t+}})\cdot \frac{\gamma w}{\cancel{\gamma w+1}}\right]}{(1-\gamma)p(\mathbf{s_{t+1}}=\mathbf{s_{t+}}\mid \mathbf{s_t},\mathbf{a_t}) + \mathbb{E}\left[(1+\gamma w)p(\mathbf{s_{t+}})\right]}$$

$$= \frac{(1-\gamma)\frac{p(\mathbf{s_{t+1}}=\mathbf{s_{t+}}\mid \mathbf{s_t},\mathbf{a_t})}{p(\mathbf{s_{t+}})} + \gamma\mathbb{E}[w]}{(1-\gamma)\frac{p(\mathbf{s_{t+1}}=\mathbf{s_{t+}}\mid \mathbf{s_t},\mathbf{a_t})}{p(\mathbf{s_{t+}})} + \gamma\mathbb{E}[w] + 1}.$$

(13)

Note that $w$ is a random variable because it depends on $\mathbf{s_{t+1}}$ and $\mathbf{a_{t+1}}$, so we take its expectation above. We can write the assignment equation for $C$ as

$$C_\theta^\pi(F=1\mid \mathbf{s_t},\mathbf{a_t},\mathbf{s_{t+}}) \leftarrow \mathbb{E}[y\mid \mathbf{s_t},\mathbf{a_t},\mathbf{s_{t+}}].$$

Noting that the function $\frac{C}{1-C}$ is strictly monotone increasing, the assignment equation is equivalent to the following assignment for the *ratio* $\frac{C_\theta^\pi(F=1\mid \mathbf{s_t},\mathbf{a_t},\mathbf{s_{t+}})}{C_\theta^\pi(F=0\mid \mathbf{s_t},\mathbf{a_t},\mathbf{s_{t+}})}$:

$$\frac{C_\theta^\pi(F=1\mid \mathbf{s_t},\mathbf{a_t},\mathbf{s_{t+}})}{C_\theta^\pi(F=0\mid \mathbf{s_t},\mathbf{a_t},\mathbf{s_{t+}})} \leftarrow \frac{\mathbb{E}[y\mid \mathbf{s_t},\mathbf{a_t},\mathbf{s_{t+}}]}{1-\mathbb{E}[y\mid \mathbf{s_t},\mathbf{a_t},\mathbf{s_{t+}}]} = (1-\gamma)\frac{p(\mathbf{s_{t+1}}=\mathbf{s_{t+}}\mid \mathbf{s_t},\mathbf{a_t})}{p(\mathbf{s_{t+}})} + \gamma\mathbb{E}[w].$$

The equality follows from substituting Eq. 13 and then simplifying. Substituting our definition of $w$, we observe that the assignment equation for off-policy C-learning is exactly the same as the assignment equation for the C-learning Bellman equation (Eq. 11):

$$\frac{C_\theta^\pi(F=1\mid \mathbf{s_t},\mathbf{a_t},\mathbf{s_{t+}})}{C_\theta^\pi(F=0\mid \mathbf{s_t},\mathbf{a_t},\mathbf{s_{t+}})} \leftarrow (1-\gamma)\frac{p(\mathbf{s_{t+1}}=\mathbf{s_{t+}}\mid \mathbf{s_t},\mathbf{a_t})}{p(\mathbf{s_{t+}})} + \gamma\left[\frac{C_\theta^\pi(F=1\mid \mathbf{s_{t+1}},\mathbf{a_{t+1}},\mathbf{s_{t+}})}{C_\theta^\pi(F=0\mid \mathbf{s_{t+1}},\mathbf{a_{t+1}},\mathbf{s_{t+}})}\right].$$

is *different* from the future state classifier used in C-learning. The reward function can then be estimated as $r_{\mathbf{s_{t+}}}(\mathbf{s_t},\mathbf{a_t}) = \frac{q_\theta(F=1\mid \mathbf{s_t},\mathbf{a_t},\mathbf{s_{t+}})}{q_\theta(F=0\mid \mathbf{s_t},\mathbf{a_t},\mathbf{s_{t+}})}$. If we learned this next state classifier, we estimate the future state density and learn goal-reaching policies by applying standard Q-learning to this reward function.

$\square$

Since the off-policy C-learning assignments are equivalent to the assignments of the C-learning Bellman equation, any convergence guarantee that applies to the later must apply to the former. Thus, Lemma 2 tells us that off-policy C-learning must also converge to the Bayes-optimal classifier. We state this final result formally:

**Corollary 3.1.** *If we use a tabular representation for the classifier, then off-policy C-learning converges to the Bayes-optimal classifier. In this case, the predicted future state density (Eq. 2) also converges to the true future state density.*

### D.3 GOAL-CONDITIONED C-LEARNING CONVERGES

In this section we prove that the version of policy improvement done by C-learning is guaranteed to improve performance. We start by noting a Bellman *optimality* equation for goal-conditioned C-learning, which indicates whether a goal-conditioned policy is optimal:

**Lemma 4** (C-learning Bellman *Optimality* Equation). *Let dynamics function $p(\mathbf{s_{t+1}} \mid \mathbf{s_t}, ca)$, and marginal distribution $p(\mathbf{s_{t+}})$ be given. If a classifier $C_\theta$ is the Bayes-optimal classifier, then it satisfies the follow identity for all states $\mathbf{s_t}$, actions $\mathbf{a_t}$, and goals $g = \mathbf{s_{t+}}$:*

$$\frac{C_\theta^\pi(F = 1 \mid \mathbf{s_t}, \mathbf{a_t}, \mathbf{s_{t+}})}{C_\theta^\pi(F = 0 \mid \mathbf{s_t}, \mathbf{a_t}, \mathbf{s_{t+}})} = (1-\gamma)\frac{p(\mathbf{s_{t+1}} = \mathbf{s_{t+}} \mid \mathbf{s_t}, \mathbf{a_t})}{p(\mathbf{s_{t+}})} + \gamma \mathbb{E}_{p(\mathbf{s_{t+1}}|\mathbf{s_t},\mathbf{a_t})}\left[\max_{\mathbf{a_{t+1}}} \frac{C_\theta^\pi(F = 1 \mid \mathbf{s_{t+1}}, \mathbf{a_{t+1}}, \mathbf{s_{t+}})}{C_\theta^\pi(F = 0 \mid \mathbf{s_{t+1}}, \mathbf{a_{t+1}}, \mathbf{s_{t+}})}\right]$$
(14)

We now apply the standard policy improvement theorem to C-learning.

**Lemma 5.** *If the estimate of the future state density is accurate, then updating the policy according to Eq. 5.2 guarantees improvement at each step.*

*Proof.* We use $\pi$ to denote the current policy and $\pi'$ to denote the policy that acts greedily w.r.t. the current density function:

$$\pi'(\mathbf{a_t} \mid \mathbf{s_t}, \mathbf{s_{t+}}) = \mathbb{1}(\mathbf{a_t} = \arg\max_a p^\pi(\mathbf{s_{t+}} \mid \mathbf{s_t}, \mathbf{a_t}))$$

The proof is quite similar to the standard policy improvement proof for Q-learning.

$$\begin{aligned}
p^\pi(\mathbf{s_{t+}} \mid \mathbf{s_t}) &= \mathbb{E}_{\pi(\mathbf{a_t}|\mathbf{s_t},\mathbf{s_{t+}})}[p^\pi(\mathbf{s_{t+}} \mid \mathbf{s_t}, \mathbf{a_t})] \\
&= \mathbb{E}_{\pi(\mathbf{a_t}|\mathbf{s_t},\mathbf{s_{t+}})}[(1-\gamma)p(\mathbf{s_{t+1}} = \mathbf{s_{t+}} \mid \mathbf{s_t}, \mathbf{a_t}) + \gamma p^\pi(\mathbf{s_{t+}} \mid \mathbf{s_{t+1}}, \mathbf{a_{t+1}})] \\
&\leq \mathbb{E}_{\pi'(\mathbf{a_t}|\mathbf{s_t},\mathbf{s_{t+}})}[(1-\gamma)p(\mathbf{s_{t+1}} = \mathbf{s_{t+}} \mid \mathbf{s_t}, \mathbf{a_t}) + \gamma p^\pi(\mathbf{s_{t+}} \mid \mathbf{s_{t+1}}, \mathbf{a_{t+1}})] \\
&= \mathbb{E}_{\pi'(\mathbf{a_t}|\mathbf{s_t},\mathbf{s_{t+}})}[(1-\gamma)p(\mathbf{s_{t+1}} = \mathbf{s_{t+}} \mid \mathbf{s_t}, \mathbf{a_t}) \\
&\quad + \mathbb{E}_{\substack{p(\mathbf{s_{t+1}}|\mathbf{s_t},\mathbf{a_t}), \\ \pi(\mathbf{a_{t+1}}|\mathbf{s_{t+1}},\mathbf{s_{t+}})}}[\gamma((1-\gamma)p(\mathbf{s_{t+1}} = \mathbf{s_{t+}} \mid \mathbf{s_{t+1}}, \mathbf{a_{t+1}}) + \gamma p^\pi(\mathbf{s_{t+}} \mid s_{t+2}, a_{t+2}))]] \\
&\leq \mathbb{E}_{\pi'(\mathbf{a_t}|\mathbf{s_t},\mathbf{s_{t+}})}[(1-\gamma)p(\mathbf{s_{t+1}} = \mathbf{s_{t+}} \mid \mathbf{s_t}, \mathbf{a_t}) \\
&\quad + \mathbb{E}_{\substack{p(\mathbf{s_{t+1}}|\mathbf{s_t},\mathbf{a_t}), \\ \pi'(\mathbf{a_{t+1}}|\mathbf{s_{t+1}},\mathbf{s_{t+}})}}[\gamma((1-\gamma)p(\mathbf{s_{t+1}} = \mathbf{s_{t+}} \mid \mathbf{s_{t+1}}, \mathbf{a_{t+1}}) + \gamma p^\pi(\mathbf{s_{t+}} \mid s_{t+2}, a_{t+2}))]] \\
&\cdots \\
&\leq p^{\pi'}(\mathbf{s_{t+}} \mid \mathbf{s_t})
\end{aligned}$$

$\square$

Taken together with the convergence of off-policy C-learning, this proof guarantees that goal-conditioned C-learning converges to the optimal goal-reaching policy (w.r.t. the functional in Eq. 5.2) in the tabular setting.

## E MIXING TD C-LEARNING WITH MC C-LEARNING

Recall that the main challenge in constructing an off-policy procedure for learning the classifier was getting samples from the future state distribution of a *new* policy. Recall that TD C-learning

(Alg. 3) uses importance weighting to estimate expectations under this new distribution, where the importance weights are computed using the learned classifier. However, this approach can result in high-variance, especially when the new policy has a future state distribution that is very different from the background distribution. In this section we describe how to decrease the variance of this importance weighting estimator at the cost of increasing bias.

The main idea is to combine TD C-learning with MC C-learning. We will modify off-policy C-learning to also use samples $\hat{p}(\mathbf{s_{t+}} \mid \mathbf{s_t}, \mathbf{a_t})$ from *previous policies* as positive examples. These samples will be sampled from trajectories in the replay buffer, in the same way that samples were generated for MC C-learning. We will use a mix of these on-policy samples (which are biased because they come from a different policy) and importance-weighted samples (which may have higher variance). Weighting the TD C-learning estimator by $\lambda$ and the MC C-learning estimator by $(1 - \lambda)$, we get the following objective:

$$\lambda \mathbb{E}_{p(\mathbf{s_{t+1}}|\mathbf{s_t},\mathbf{a_t}),p(\mathbf{s_{t+}})}[(1-\gamma)\log C(F=1 \mid \mathbf{s_t}, \mathbf{a_t}, \mathbf{s_{t+1}}) + \log C(F=0 \mid \mathbf{s_t}, \mathbf{a_t}, \mathbf{s_{t+}})$$
$$+ \gamma w \log C(F=1 \mid \mathbf{s_t}, \mathbf{a_t}, \mathbf{s_{t+}})$$
$$+ (1-\lambda)\mathbb{E}_{\hat{p}(\widehat{\mathbf{s_{t+}}}|\mathbf{s_t},\mathbf{a_t}),p(\mathbf{s_{t+}})}[\log C(F=1 \mid \mathbf{s_t}, \mathbf{a_t}, \widehat{\mathbf{s_{t+}}}) + \log C(F=0 \mid \mathbf{s_t}, \mathbf{a_t}, \mathbf{s_{t+}})]$$

This method is surprisingly easy to implement. Given a batch of $B$ transitions $(\mathbf{s_t}, \mathbf{a_t})$, we label $\frac{\lambda}{2}B$ with the next state $\mathbf{s_{t+1}}$, $\frac{1}{2}B$ with a random state $\mathbf{s_{t+}} \sim p(\mathbf{s_{t+}})$, and $\frac{1-\lambda}{2}B$ with a state sampled from the empirical future state distribution $\hat{p}(\mathbf{s_{t+}} \mid \mathbf{s_t}, \mathbf{a_t})$. To make sure that each term in the loss above receives the correct weight, we scale each of the terms by the inverse sampling probability:

(Next states): $\quad \dfrac{2}{\cancel{\lambda}B}(\cancel{\lambda}(1-\gamma)\log C(F=1 \mid \mathbf{s_t}, \mathbf{a_t}, \mathbf{s_{t+1}})$

(Random states): $\quad \dfrac{2}{B}\left((\cancel{\lambda}+1-\cancel{\lambda})\log C(F=0 \mid \mathbf{s_t}, \mathbf{a_t}, \mathbf{s_{t+}}) + \lambda\gamma w \log C(F=1 \mid \mathbf{s_t}, \mathbf{a_t}, \mathbf{s_{t+}})\right)$

(Future states): $\quad \dfrac{2}{\cancel{(1-\lambda)}B}\cancel{(1-\lambda)}\log C(F=1 \mid \mathbf{s_t}, \mathbf{a_t}, \mathbf{s_{t+}})$

Without loss of generality, we scale each term by $\frac{B}{2}$. Since each of these terms is a cross entropy loss, we can simply implement this loss as a weighted cross entropy loss, where the weights and labels are given in the table below.

| | Fraction of batch | Label | Weight |
|---|---|---|---|
| Next states | $\frac{\lambda}{2}$ | 1 | $1-\gamma$ |
| Future states | $\frac{1-\lambda}{2}$ | 1 | 1 |
| Random states | $\frac{1}{2}$ | $\frac{\lambda\gamma w}{1+\lambda\gamma w}$ | $(1+\lambda\gamma w)$ |

On many tasks, we observed that this approach performed no differently than TD C-learning. However, we found this strategy to be crucial for learning some of the sawyer manipulation tasks. In our experiments we used $\lambda = 0.6$ for the Sawyer Push and Sawyer Drawer tasks, and used $\lambda = 1$ (i.e., pure TD C-learning) for all other tasks.

## F  ADDITIONAL EXPERIMENTS

**Comparing C-learning and C-learning for Future State Density Estimation**   Fig. 5 shows the results of our comparison of C-learning and Q-learning on the "continuous gridworld" environment, in both the on-policy and off-policy setting. In both settings, off-policy C-learning achieves lower error than Q-learning. As expected, Monte Carlo C-learning performs well in the on-policy setting, but poorly in the off-policy setting, motivating the use of off-policy C-learning.

**Additional Results on Predicting the Goal Sampling Ratio**   To further test Hypothesis 2, we repeated the experiment from Fig. 1b across a range of values for $\gamma$. As shown in Fig. 6, our Hypothesis accurately predicts the optimal goal sampling ratio across a wide range of values for $\gamma$.

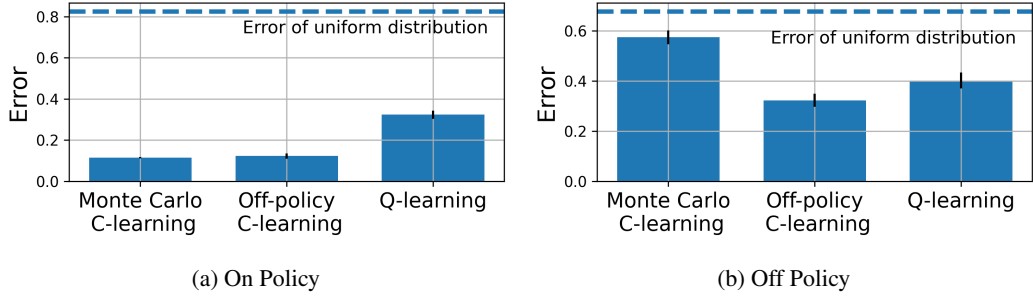

|                     |                    |
| ------------------- | ------------------ |
| (a) On Policy       | (b) Off Policy     |

Figure 5: We use C-learning and Q-learning to predict the future state distribution. *(Right)* In the on-policy setting, both the Monte Carlo and TD versions of C-learning achieve significantly lower error than Q-learning. *(Right)* In the off-policy setting, the TD version of C-learning achieves lower error than Q-learning, while Monte Carlo C-learning performs poorly, as expected.

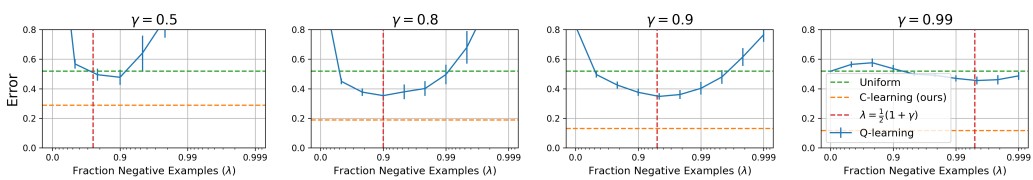

Figure 6: The performance of Q-learning (blue line) is sensitive to the relabeling ratio. Our analysis accurately predicts that the optimal relabeling ratio is approximately $\lambda = \frac{1}{2}(1 + \gamma)$. Our method, C-learning, does not require tuning this ratio, and outperforms Q-learning, even with the relabeling ratio for Q-learning is optimally chosen.

## G  EXPERIMENTAL DETAILS

### G.1  "CONTINUOUS GRIDWORLD" EXPERIMENTS

**The "Continuous Gridworld" Environment**  Our first set of experiments aimed to compare the predictions of Q-learning and C-learning to the true future state density. We carefully chose the environment to conduct this experiment. We want it to have stochastic dynamics and a continuous state space, so that the true Q function for the indicator reward is 0. On the other hand, to evaluate our hypotheses, we want to be able to analytically compute the true future state density. Thus, we use a modified $5 \times 5$ gridworld environment where the agent observes a noisy version of the current state. Precisely, when the agent is in position $(i, j)$, the agent observes $(i + \epsilon_i, j + \epsilon_j)$ where $\epsilon_i, \epsilon_j \sim \text{Unif}[-0.5, 0.5]$. Note that the observation uniquely identifies the agent's position, so there is no partial observability. We can analytically compute the exact future state density function by first computing the future state density of the underlying gridworld and noting that the density is uniform within each cell (see Appendix B). We generated a tabular policy by sampling from a Dirichlet(1) distribution, and sampled 100 trajectories of length 100 from this policy.

**On-Policy Experiment**  We compare C-learning (Algorithms 1 and 2) to Q-learning in the on-policy setting, where we aim to estimate the future state density function of the same policy that collected the dataset. This experiment aims to answer whether C-learning and Q-learning solve the future state density estimation problem (Def. 2). Each of the algorithms used a 2 layer neural network with a hidden layer of size 32, optimized for 1000 iterations using the Adam optimizer with a learning rate of 3e-3 and a batch size of 256. C-learning and Q-learning with hindsight relabeling differ only in their loss functions, and the fact that C-learning acquires a classifier while Q-learning predicts a continuous Q-value. After training with each of the algorithms, we extracted the estimated future state distribution $f_\theta^\pi(\mathbf{s_{t+}} \mid \mathbf{s_t}, \mathbf{a_t})$ using Eq. 2. We also normalized the predicted distributions $f_\theta^\pi(\mathbf{s_{t+}} \mid \mathbf{s_t}, \mathbf{a_t})$ to sum to 1 (i.e., $\sum_{s_{t+}} f_\theta^\pi(\mathbf{s_{t+}} = s_{t+} \mid \mathbf{s_t}, \mathbf{a_t}) = 1 \; \forall \mathbf{s_t}, \mathbf{a_t}$). For evaluation, we computed the KL divergence with the true future state distribution. We show results in Fig. 5a.

Figure 7: **Predicting the Future with C-Learning**: We ran the experiment from Fig. 2 on three locomotions tasks.

**Off-Policy Experiment**  We use the same "continuous gridworld" environment to compare C-learning to Q-learning in the off-policy setting, where we want to estimate the future state distribution of a policy that is different from the behavior policy. Recall that the motivation for deriving the bootstrapping version of the classifier learning algorithm was precisely to handle this setting. To conduct this experiment, we generated two tabular policies by sampling from a Dirichlet(1) distribution, using one policy for data collection and the other for evaluation. We show results in Fig. 5b.

**Experiment Testing Hypothesis 1 (Fig. 1a)**  For this experiment, we found that using a slightly larger network improved the results of both methods, so we increased hidden layer size from 32 to 256.

**Experiment Testing Hypothesis 2 (Fig. 1b, Fig. 6)**  To test this hypothesis, we used the Q-learning objective in Eq. 10, where $\lambda$ represents probability of sampling a random state as $\mathbf{s_{t+}}$. Following prior work (Fu et al., 2019), we reweight the loss function instead of changing the actual sampling probabilities, thereby avoiding additional sampling error. We ran this experiment in the on-policy setting, using 5 random seeds for each value of $\lambda$. For each trial, we normalized the predicted density function to sum to 1 and then computed the KL divergence with the true future state density function.

### G.2   PREDICTION EXPERIMENTS ON MUJOCO LOCOMOTION TASKS

In this section we provide details for the prediction experiment discussed in Sec. 6 and plotted in Fig. 2 and Fig. 7.

We first describe how we ran the experiment computing the prediction error in Fig. 7a- 7c. For these experiments, we used the "expert" data provided for each task in Fu et al. (2020). We split these trajectories into train (80%) and test (20%) splits. All methods (MC C-learning, TD C-learning, and the 1-step dynamics model) used the same architecture (one hidden layer of size 256 with ReLU activation), but C-learning output a binary prediction whereas the 1-step dynamics model output a vector with the same dimension as the observations. While we could have used larger networks and likely learned more accurate models, the aim of this experiment is to compare the relative performance of C-learning and the 1-step dynamics model when they use similarly-expressive networks. We trained MC C-learning and the 1-step dynamics model for 3e4 batches and trained the TD C-learning model for just 3e3 batches (we observed overfitting after that point). For TD C-learning, we clipped $w$ to lie in $[0, 20]$. To evaluate each model, we randomly sampled a 1000 state-action pairs from the validation set and computed the average MSE with the empirical expected future state. We computed the empirical expected future state by taking a geometric-weighted average of the next 100 time steps. We computed the predictions from the 1-step model by unrolling the model for 100 time steps and taking the geometric-weighted average of these predictions. To compute predictions from the classifier, we evaluate the classifier at 1000 randomly-sampled state-action pairs (taken from the same dataset), convert these predictions to importance weights using Eq. 2, normalize the importance weights to sum to 1, and then take the weighted average of the randomly-sampled states.

To visualize the predictions from C-learning (Fig. 2b), we had to modify these environments to include the global $X$ coordinate of the agent in the observation. We learned policies for solving these modified tasks by running the SAC (Haarnoja et al., 2018) implementation in (Guadarrama et al., 2018) using the default hyparameters. We created a dataset of 1e5 transitions for each environment. We choose the maximum horizon for each task based on when the agent ran out of the frame (200 steps for `HalfCheetah-v2`, 1000 steps for `Hopper-v2`, 1000 steps for `Walker2d-v2`, and 300 steps for `Ant-v2`). We then trained TD C-learning on each of these datasets. The hyperparameters were the same as before, except that we trained for 3e4 batches. We evaluated the classifier

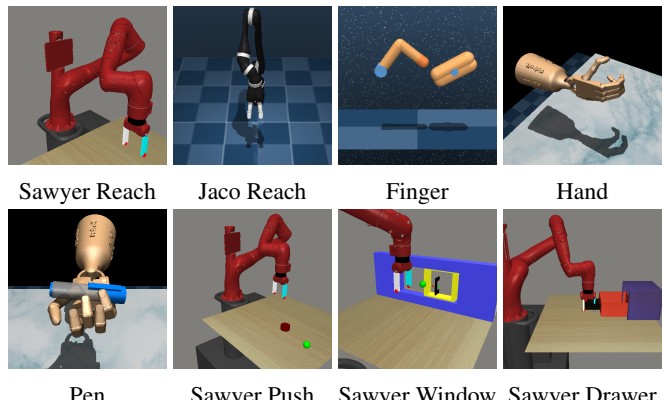

Figure 8: Continuous Control Environments

at 1000 randomly-sampled state-action pairs taken from a separate validation set, computed the importance weights as before, and then took the weighted average of the rendered images of each of these 1000 randomly-sampled states. We normalized the resulting images to be in [0, 255].

### G.3 GOAL-CONDITIONED RL EXPERIMENTS

In this section we describe the continuous control tasks used in our experiments (shown in Fig. 8), as well as the hyperparameters used in our implementation of goal-conditioned C-learning. One important detail is that we used a subset of the state coordinates as the goal in many tasks. For example, in the Jaco Reach task, we used just the joint angles, not the joint velocities, as the goal. When C-learning is only conditioned on a subset of the state coordinates, it estimates the *marginal* future state distribution over just those coordinates. Unless otherwise mentioned, environments used the default episode length. Code will be released.

- **Jaco Reach** This task is based on the manipulation environment in Tassa et al. (2020). We sampled goals by resetting the environment twice, using the state after the first reset as the goal. For this task we used just the position, not the velocity, of the joint angles as the goal.

- **Finger** This task is based on the `Finger` task in Tassa et al. (2018). We sampled goals by resetting the environment twice, using the state after the first reset as the goal. For this task we used the position of the spinner as the goal.

- **Hand** This task is a modified version of the `door-human-v0` task in Fu et al. (2020), but modified to remove the door, so just the hand remains. We sampled goals by taking 50 random actions, recording the current state as the goal, and then resetting the environment. This task used the entire state as the goal. Episodes had length 50.

- **Pen** This task is based on the `pen-human-v0` task in Fu et al. (2020). We sampled goals by randomly choosing a state from a dataset of human demonstrations provided in Fu et al. (2020). For this task we used the position (but not orientation) of the pen as the goal.

- **Sawyer Reach** This task is based on the `SawyerReachXYZEnv-v0` environment from Yu et al. (2020). We used the default goal sampling method, and used the end effector position as the goal. Episodes had length 50.

- **Sawyer Push** This task is based on the `SawyerReachPushPickPlaceEnv-v0` environment from Yu et al. (2020). We sampled goals uniformly from $\text{Unif}([-0.1, 0.1], [0.5, 0.9], [0.015, 0.015])$, using the same goal for the arm and puck. Episodes had length 150.

- **Sawyer Window** This task is based on the `SawyerWindowCloseEnv-v0` environment from Yu et al. (2020). We randomly sampled the initial state and goal state uniformly from the possible positions of the window inside the frame. The goal for the arm was set to be the same as the goal for the window. Episodes had length 150.

|  | C-learning | TD3 + Next State Relabeling |
|---|---|---|
| Jaco Reach | 0.9 | 0.99 |
| Finger | 0.99 | 0.99 |
| Hand | 0.3 | 0.8 |
| Pen | 0.7 | 0.99 |
| Sawyer Reach | 0.99 | 0.99 |
| Sawyer Push | 0.99 | 0.99 |
| Sawyer Window | 0.99 | 0.99 |
| Sawyer Drawer | 0.99 | 0.99 |

Table 1: Values of the discount $\gamma$ for C-learning and TD3 with Next State Relabeling

- **Sawyer Drawer** This task is based on the `SawyerDrawerOpenEnv-v0` environment from Yu et al. (2020). We randomly sampled the initial state and goal state uniformly from the feasible positions of the drawer. The goal for the arm was set to be the same as the goal for the window. Episodes had length 150.

Our implementation of C-learning is based on the TD3 implementation in Guadarrama et al. (2018). We learned a stochastic policy, but (as expected) found that all methods converged to a deterministic policy. We list the hyperparameter below, noting that almost all were taken without modification from the SAC implementation in Guadarrama et al. (2018) (the SAC implementation has much more reasonable hyperparameters):

- Actor network: 2 fully-connected layers of size 256 with ReLU activations.

- Critic network: 2 fully-connected layers of size 256 with ReLU activations.

- Initial data collection steps: 1e4

- Replay buffer size: 1e6

- Target network updates: Polyak averaging at every iteration with $\tau = 0.005$

- Batch size: 256. We tried tuning this but found no effect.

- Optimizer: Adam with a learning rate of 3e-4 and default values for $\beta$

- Data collection: We collect one transition every one gradient step.

When training the policy (Eq. **??**), we used the same goals that we sampled for the classifier (Eq. **??**), with 50% being the immediate next state and 50% being random states. The most sensitive hyperparameter was the discount, $\gamma$. We therefore tuned $\gamma$ for both C-learning and the most competitive baseline, TD3 with next state relabeling (Lin et al., 2019). The tuned values for $\gamma$ are shown in Table 1. We used $\gamma = 0.99$ for the other baselines.

In our implementation of C-learning, we found that values for the importance weight $w$ became quite larger, likely because the classifier was making overconfident predictions. We therefore clipped the values of $w$ to be in $[0, 2]$ for most tasks, though later ablation experiments found that the precise value of the upper bound had little effect. We used a range of $[0, 50]$ for the finger and Sawyer tasks. Further analysis of the importance weight revealed that it effectively corresponds to the planning horizon; clipping the importance weight corresponds to ignoring the possibility of reaching goals beyond some horizon. We therefore suggest that $1/(1 - \gamma)$ is a reasonable heuristic for choosing the maximum value of $w$.

## H  ANALYTIC EXAMPLES OF Q-LEARNING FAILURES

In this section we describe a few simple MDPs where Q-learning with hindsight relabeling fails to learn the true future state distribution. While we describe both examples as discrete state MDPs, we assume that observations are continuous and noisy, as described in Appendix G

## H.1 EXAMPLE 1

The first example is a Markov process with $n$ states. Each state deterministically transitions to itself. We will examine the one state, $s_1$ (all other states are symmetric). If the agent starts at $s_1$, it will remain at $s_1$ at every future time step, so $p(\mathbf{s_{t+}} \mid \mathbf{s_t} = s_1) = \mathbb{1}(\mathbf{s_{t+}} = s_1)$.

We use $\lambda$, defined above, to denote the relabeling fraction. The only transition including state $s_1$ is $(s_t = s_1, s_{t+1} = s_1)$. We aim to determine $Q = Q(\mathbf{s_t} = s_1, \mathbf{s_{t+}} = s_1, \mathbf{s_{t+}} = s_1)$. There are two possible ways we can observe the transition $(s_t = s_1, \mathbf{s_{t+}} = s_1, \mathbf{s_{t+}} = s_1)$. First, with probability $1 - \lambda$ we sample the next state as $\mathbf{s_{t+}}$, and the TD target is 1. Second, with probability $\lambda$ we sample a random state with TD target $\gamma Q$, and with probability $1/n$ this random state is $s_1$. Thus, conditioned on $\mathbf{s_{t+}} = s_1$, the Bellman equation is

$$Q = \begin{cases} 1 & \text{w.p. } 1 - \lambda \\ \gamma Q & \text{w.p. } \frac{\lambda}{n} \end{cases}.$$

Solving for $Q$, we get $Q^* = \frac{1-\gamma}{1-\frac{\lambda\gamma}{n}}$. This Q function is clearly different from the true future state distribution, $p(\mathbf{s_{t+}} \mid \mathbf{s_t} = s_1) = \mathbb{1}(\mathbf{s_{t+}} = s_1)$. First, since $\frac{\lambda}{n} < 1$, the Q function is greater than one ($Q^* > 1$), but a density function over a discrete set of states cannot be greater than one. Second, even if we normalized this Q function to be less than one, we observe that the Q function depends on the discount ($\gamma$), the relabeling ratio ($\lambda$), and the number of states ($n$). However, the true future state distribution has no such dependence. Thus, we conclude that even scaling the optimal Q function by a constant (even one that depends on $\gamma$!) would not yield the true future state distribution.

## H.2 EXAMPLE 2

Our second example is a stochastic Markov process with two states, $s_1$ and $s_2$. The transition probabilities are

$$p(\mathbf{s_{t+1}} \mid \mathbf{s_t} = s_1) = \begin{cases} \frac{1}{2} & \text{if } \mathbf{s_{t+1}} = s_1 \\ \frac{1}{2} & \text{if } \mathbf{s_{t+1}} = s_2 \end{cases}, \qquad p(\mathbf{s_{t+1}} \mid \mathbf{s_t} = s_2) = \mathbb{1}(\mathbf{s_{t+1}} = s_2)$$

We assume that we have observed each transition once. To simplify notation, we will use $Q_{ij} = Q(\mathbf{s_t} = s_i, \mathbf{s_{t+}} = s_j)$. There are three ways to end up with $Q_{11}$ on the LHS of a Bellman equation: (1) the next state is $s_1$ and we sample the next state as the goal, (2) the next state is $s_1$ and we sample a random state as the goal, and (3) the next state is $s_2$ and we sample a random state as the goal.

$$Q_{11} = \begin{cases} 1 & \text{w.p. } 1 - \lambda \\ \gamma Q_{11} & \text{w.p. } \frac{\lambda}{2} \\ \gamma Q_{21} & \text{w.p. } \frac{\lambda}{2} \end{cases}.$$

We can likewise observe $Q_{12}$ on the LHS in the same three ways: (1) the next state is $s_2$ and we sample the next state as the goal, (2) the next state is $s_1$ and we sample a random state as the goal, and (3) the next state is $s_2$ and we sample a random state as the goal.

$$Q_{12} = \begin{cases} 1 & \text{w.p. } 1 - \lambda \\ \gamma Q_{12} & \text{w.p. } \frac{\lambda}{2} \\ \gamma Q_{22} & \text{w.p. } \frac{\lambda}{2} \end{cases}. \tag{15}$$

We can only observe $Q_{21}$ on the LHS in one way: the next state is $s_2$ and we sample we sample a random state as the goal:

$$Q_{21} = \begin{cases} \gamma Q_{21} & \text{w.p. } 1 \end{cases}. \tag{16}$$

We can observe $Q_{22}$ on the LHS in two ways: (1) the next state is $s_2$ and we sample the next state as the goal, (2) the next state is $s_2$ and we sample a random state as the goal:

$$Q_{22} = \begin{cases} 1 & \text{w.p. } \frac{1-\lambda}{1-\lambda+\frac{\lambda}{2}} \\ \gamma Q_{22} & \text{w.p. } \frac{\frac{\lambda}{2}}{1-\lambda+\frac{\lambda}{2}} \end{cases}. \tag{17}$$

To solve these equations, we immediately note that the only solution to Eq. 16 is $Q_{21} = 0$. Intuitively this makes sense, as there is zero probability of transition $s_2 \rightarrow s_1$. We can also solve Eq. 17 directly:

$$(1 - \frac{\gamma\lambda}{2-\lambda})Q_{22} = \frac{2-2\lambda}{2-\lambda} \implies \frac{2-\lambda+\gamma\lambda}{2-\lambda}Q_{22} = \frac{2-2\lambda}{2-\lambda} \implies Q_{22} = \frac{2-2\lambda}{2-\lambda+\gamma\lambda}.$$

Next, we solve Eq. 15. We start by rearranging terms and substituting our solution to $Q_{22}$:

$$(1 - \frac{\gamma\lambda}{2})Q_{12} = 1 - \lambda + \frac{\gamma\lambda}{2}Q_{22} = 1 - \lambda + \frac{\gamma\lambda}{2}\frac{2-2\lambda}{2-\lambda+\gamma\lambda} = (1-\lambda)\frac{2-\lambda+\gamma\lambda+\gamma\lambda}{2-\lambda+\gamma\lambda}$$

Rearranging terms, we obtain the following:

$$Q_{12} = \frac{2(1-\lambda)(2-\lambda+2\gamma\lambda)}{(2-\gamma\lambda)(2-\lambda+\gamma\lambda)}$$

Finally, we solve for $Q_{11}$, recalling that our solution $Q_{21} = 0$:

$$(1 - \frac{\gamma\lambda}{2})Q_{11} = 1 - \lambda + \frac{\gamma\lambda}{2}Q_{21}^{\cancel{\quad}0} \implies Q_{11} = \frac{2-2\lambda}{2-\gamma\lambda}$$

To summarize, Q-learning with hindsight relabeling obtains the following Q values:

$$Q_{11} = \frac{2(1-\lambda)}{2-\gamma\lambda}, Q_{12} = \frac{2(1-\lambda)(2-\lambda+2\gamma\lambda)}{(2-\gamma\lambda)(2-\lambda+\gamma\lambda)}, Q_{21} = 0, Q_{22} = \frac{2-2\lambda}{2-\lambda+\gamma\lambda}.$$

For comparison, we compute $p(\mathbf{s_{t+}} = s_1 \mid s_1)$, the probability that the agent remains in state $s_1$ in the future. The probability that the agent is in $s_1$ at time step $\Delta$ is $1/2^\Delta$, so the total, discounted probability is

$$p(\mathbf{s_{t+}} = s_1 \mid s_1) = (1-\gamma)(1 + \frac{1}{2}\gamma + \frac{1}{4}\gamma^2 + \cdots)$$

$$= (1-\gamma)\sum_{\Delta=0}^{\infty}\left(\frac{\gamma}{2}\right)^\Delta$$

$$= \frac{1-\gamma}{1-\gamma/2} = \frac{2-2\gamma}{2-\gamma}.$$

Thus, in general, the Q value $Q_{11}$ does not equal the future state distribution. One might imagine that Q-learning acquires Q values that are only accurate up to scale, so we should consider the normalized prediction:

$$\frac{Q_{11}}{Q_{11}+Q_{12}} = \frac{1}{1 + \frac{2-\lambda+2\gamma\lambda}{2-\lambda+\gamma\lambda}} = \frac{2-\lambda+\gamma\lambda}{4-2\lambda+3\gamma\lambda}$$

However, this normalized Q-learning prediction is also different from the true future state distribution.

# I   PREDICTIONS FROM C-LEARNING

Figures 9 and 10 visualizes additional predictions from the C-learning model in Sec. 6. In each image, the top half shows the current state and the bottom half shows the predicted expected future state. Animations of these results can be found on the project website.

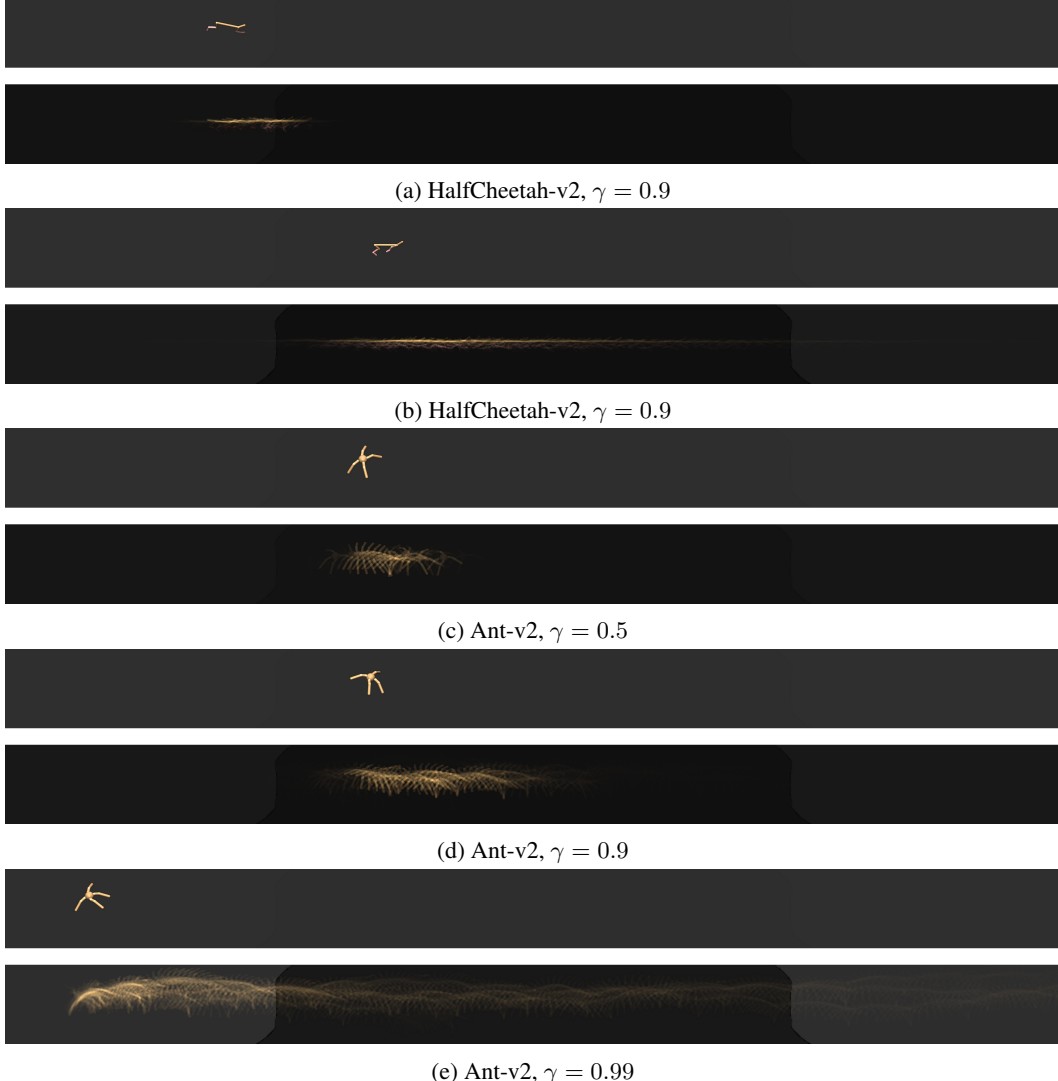

(a) HalfCheetah-v2, $\gamma = 0.9$

(b) HalfCheetah-v2, $\gamma = 0.9$

(c) Ant-v2, $\gamma = 0.5$

(d) Ant-v2, $\gamma = 0.9$

(e) Ant-v2, $\gamma = 0.99$

Figure 9: Predictions from C-learning

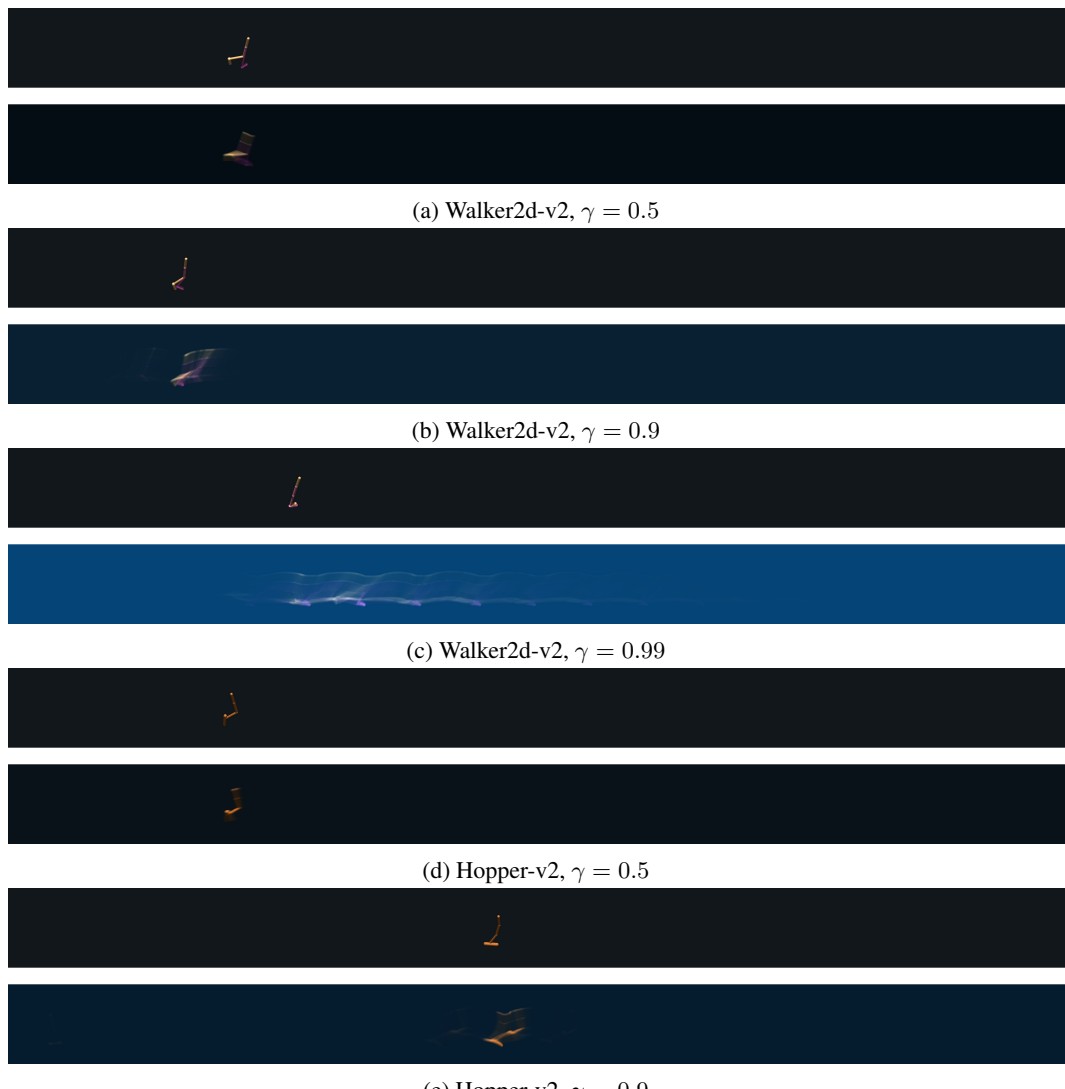

(a) Walker2d-v2, $\gamma = 0.5$

(b) Walker2d-v2, $\gamma = 0.9$

(c) Walker2d-v2, $\gamma = 0.99$

(d) Hopper-v2, $\gamma = 0.5$

(e) Hopper-v2, $\gamma = 0.9$

Figure 10: More Predictions from C-learning

