# OpenReview forum: "C-Learning: Learning to Achieve Goals via Recursive Classification"
_ICLR.cc/2021/Conference — ICLR 2021 Poster_

### Official Review · AnonReviewer2 · 2020-10-20
**Good idea, but confusing manuscripts.**

**Rating:** 6
**Confidence:** 2

**Review:**

The authors propose a new algorithm, called C-learning, which tackles goal-conditioned reinforcement learning problems. Specifically, the algorithm converts the future density estimation problem, which goal-conditioned Q learning is inherently performing, to a classification problem. The experiments showed that the modification allows a more precise density estimation than Q-learning, and in turn, a good final policy.

Overall, I like the general idea to use classification as a tool for estimating the future density function. Especially, the idea is valuable in that it allows a better understanding of prior Q-learning based approaches in choosing a sensitive hyperparameter. However, the manuscript can be enhanced much by adapting more precise notations and adding more explanations on equations:
* $p$ is highly overloaded; it is used to represent future conditional state density function and marginal state density function for both on-policy and off-policy, and transition dynamics.
* Also, it would be important to notate $\pi$ in most of the parts, including $p$, $Q$, and $C$ (unless it is very obvious). Especially, the current notation is very confusing when the off-policy algorithm is introduced.
* Related to this concern, I am not fully convinced of the off-line algorithm due to the marginal future state distribution $p(s_{t+})$. Doesn’t it supposed to be $p^{\pi_{\text{eval}}}(s_{t+})$, and therefore, does the marginal distribution also need to be adjusted as $p(s_{t+}|s_{t+1},a_{t+1})$?
* It would be also helpful if the full derivation for Equation (6) is included in the main manuscript.

-- After rebuttal

I've read the authors' feedbacks and other reviewers' comments. My major concern was the clarity of the manuscript as other reviewers mentioned, and I believe the concern has been resolved during the rebuttal period. I adjusted my ratings representing that.

---

> ### Author Response · Authors · 2020-11-14
> **Response to R2**
>
> Thanks for the review, and for the detailed suggestions for improvement. If we understand correctly, the main concern was about the clarity of the paper. We believe that we have incorporated all the writing suggestions (details below), thereby addressing this concern. We welcome additional suggestions on improving clarity, and kindly ask the reviewer to revisit the review in light of these changes
>
> **Notation**: We have revised the paper to take into account all the writing suggestions:
> * Revised the notation for $p$ using the alternative suggested by R5
> * Added superscripts to $Q$, $f$, and $C$ to indicate which policies these functions correspond to.
> * We have added a sentence after Eq. 6 explaining how it is derived. (Let us know if this isn't clear!)
>
> We have also incorporated the writing suggestions of other reviewers. We believe that these address the concerns about clarity, but would welcome additional suggestions!
>
> **TD C-learning is an off-policy algorithm**, meaning that it can learn from experience collected from a different policy. However, _off-policy_ does not mean fully _offline_: in our goal-conditioned RL experiments using TD C-learning, we do periodically collect new experience using the learned policy. The review correctly notes that the marginal distribution p(s) therefore changes throughout learning, but this does not affect the correctness of TD C-learning. We have updated the paper to include this discussion right before the "Algorithm Summary" in Section 5.2.

---

> > ### Author Response · Authors · 2020-11-20
> > **Have the (significant) paper revisions addressed the reviewer's concerns?**
> >
> > With the rebuttal deadline just a few days away, we wanted to follow up with the reviewer to confirm that the clarifications and paper revisions discussed above have addressed all the concerns raised in the review. We emphasize that we have incorporated all the writing suggestions from all reviewers in the revised manuscript. We kindly ask the reviewer to let us know if they have any outstanding questions or concerns so that we can address these.

---

### Official Review · AnonReviewer3 · 2020-10-28
**well written and clear**

**Rating:** 8
**Confidence:** 3

**Review:**

Summary:
This work presents a goal-conditioned RL, which estimates probability density using a classifier.

Strengths:
+ The problem is well explained, the logical structure seems adequate.
+ The paper is well written and clear.
+ The approach technically sounds and mathematically well-formulated.

Weaknesses:
- Although the reported evaluation results are competitive to baselines, it would have been even stronger if the performance is substantially improved. Do you have any insight on how better results can be achieved?

---

> ### Author Response · Authors · 2020-11-14
> **Response to R3**
>
> Thanks for the review! As noted in the general comment above, we have updated the paper to include results on considerably more complex manipulation tasks. As shown in Fig 3 (updated), our method performs substantially better than baselines on these more complex tasks. One fact that contributed to the success of our method on some (but not all) of these task was to use a mix of TD C-learning and MC C-learning, which we describe in Appendix E. Please refer to the project website for videos of the policies learned on these tasks: https://c-learning-anonymous.github.io/
>
> We welcome additional suggestions on how to improve the paper and method!

---

### Official Review · AnonReviewer1 · 2020-10-29
**Interesting and relevant to UOM paper but relevance and connection is not discussed.**

**Rating:** 7
**Confidence:** 5

**Review:**

This paper studies a problem of predicting future state distribution in an MDP. The approach taken is an indirect approach which first predicts whether an observation comes from the future and transfers this binary prediction via Bayes rule to predictions over future states.

Conceptually the paper's appears novel but a previous similar work was very relevant. In UOM paper, Definition 1 is almost exactly the same as the discounted state occupancy function defined the UOM paper, just an extra normalization constant (1-\gamma). Apparently at this core the idea is not novel. However, the paper's development of casting the prediction problem into two stages (first binary prediction) and using Bayesian to transfer into the future state distribution is novel. The connection of UOM paper is interesting but not discussed unfortunately. In the UOM paper, the authors there appear to focus on reward-less MDPs, where you can generate/compute the value function given a reward function on the fly. Here the paper focuses on more on the estimation of the the discounted state occupancy function, -- this is my interpretation.

Paragraph "In discrete state spaces":
introducing this reward: again this is just another interpretation of the discounted state occupancy function in UOM (equation 2).

Remark 1:
I don't understand the point of Remark 1. do you mean for a continuous-state problems, the prob of reaching some particular state is zero? Isn't that that obvious? What is the point here?

Hypothesis 1 and 2: I don't know whether they make sense. The paper relies on the experiments to try to make sense of it. I'm not sure this is a sound approach. Why testing these hypotheses is interesting at the first place? At least this isn't clear from the paper.


UOM paper (not cited):
https://papers.nips.cc/paper/5590-universal-option-models

I've read the authors' feedbacks and other reviewers' comments. R5's main concerns are the clarity and the motivation of Bayessian classifier and off-policy learning. That should have been resolved from authors' feedbacks.

---

> ### Author Response · Authors · 2020-11-14
> **Response to R1**
>
> Thanks for the review! We believe that there may be some misunderstanding of the relationship between our paper and UOM; we aim to clarify this misunderstanding below. Both our work and UOM propose methods for predicting the future state distribution. Our work differs from UOM by describing how to learn goal-conditioned policies based on these predictions. We emphasize that our method learns these goal-conditioned policies without requiring a manually-specified reward function.
>
> Another important detail is that UOM looks at predicting the future in discrete spaces, whereas our work is applicable to both discrete and continuous state spaces. The aim of Remark 1 is to argue that in continuous environments we should think about density functions rather than probabilities: the _probability_ of reaching any particular state is 0. For example, the RHS of Eq 1 in UOM would be 0 in these sorts of environments. We have revised the paper to include a citation and discussion of UOM in the related work section. We ask the reviewer to revisit the review in light of this clarification.
>
> We have also revised the paper to clarify the aim of the hypotheses in the paper. Hypothesis 1 predicts that Q-learning with hindsight relabeling will underestimate the density function, while Hypothesis 2 predicts how to optimally sample goals for hindsight relabeling. Both hypotheses are testable predictions made by our theory. Our experiments confirming these hypotheses provide evidence that our theory is correct. Hypothesis 2 is of additional interest because prior work has shown that choosing the goal-sampling ratio is (empirically) is challenging [Zhao 19, Pong 18, Andrychowicz 17]. Our result provides a theoretically-justified and empirically-verified way to automatically choose this sensitive hyperparameter. We have incorporated this discussion in the main text, after the statements of each of the hypotheses.

---

> > ### Author Response · Authors · 2020-11-20
> > **Have the revisions and discussion of UOM addressed the reviewer's concerns?**
> >
> > With the rebuttal deadline just a few days away, we wanted to follow up with the reviewer to confirm that the paper revision (which includes a new discussion of UOM) and the discussion above addresses all the concerns raised in the review. We kindly ask the reviewer to let us know if they have any outstanding questions or concerns so that we can address these.

---

### Official Review · AnonReviewer4 · 2020-11-05
**Building probability density of reaching a future observation via contrastive classification**

**Rating:** 7
**Confidence:** 3

**Review:**

This paper explores learning a classifier to predict if a given state/observation will be reached in the future from the current state and action pair. Using this classifier, the paper is able to create a probability density function that could be conditioned on reaching a goal state. This idea is interesting and could leverage large scale self-supervised learning to build the future state probability density function. I have some concerns which are listed below.

1. How does the method work on low dimensional action spaces where the probability of reaching a future state from current state and action is usually higher?

2. How does the method work on high dimensional tasks that have discrete-continuous dynamics due to multiple contacts, such as the pen task. The probability of reaching a future state from the current state + action is very low and the experiments show only training on a human demonstrated dataset. Can the policy be used to control the hand to reach a target pen position very different from the human demonstration?

---

> ### Author Response · Authors · 2020-11-14
> **Response to R4**
>
> Thank you for the review. We'll respond to the two questions raised in the review:
> 1. C-Learning is applicable to a broad range of tasks, ranging from those where reaching future states is quite easy (like the hand environment) to those where reaching future states is quite difficult (like the pushing task). Our experiments demonstrate that C-learning works well on both types of tasks, and substantially outperforms prior methods on the difficult tasks. For easy tasks, we found that using a smaller discount factor $\gamma$ provided better results for both our method and baselines (see Table 1 in Appendix G3).
> 2. There might be some confusion here: our goal-conditioned RL experiments (Fig. 3) do not use human demonstrations for training. The only way we use human demonstrations is for defining the goal distribution on the Pen task (this is the only task where this is done). On all other tasks (including all the Sawyer manipulation tasks), the goal distribution was fixed to be uniform over the workspace. Experiments on these tasks demonstrate that C-learning successfully learns to reach a wide range of goals.
>
> We encourage the reviewer to check out the new results on the Sawyer manipulation tasks in Fig 3 and visualized on the project website: https://c-learning-anonymous.github.io/

---

> > ### Comment · AnonReviewer4 · 2020-11-20
> > **clarification helps**
> >
> > Thank you for clarifying some confusion I had.

---

### Official Review · AnonReviewer5 · 2020-11-06
**Official Blind Review #5**

**Rating:** 4
**Confidence:** 4

**Review:**

[summary]
This paper studies to predict future state density function by using an indirectly method via classification. The main idea is to sample the future state from two sources: 1) from replay buffer, 2) the actual next state in the trajectory (in off policy setting we only need the next state) and then use a classifier to distinguish them. By Bayesian rule we can recalculate the conditional density function by the ratio of the classifier. The paper compare this method with several baseline and find that they can predict the conditional density function very close to reality.

[originality]
I really like the idea and the method seems very interesting and novel to me. However, the main concern is the motivation of the classifier. It seems to me that we can classify the true $s_{t,+}$ with any source of distribution of $p(s_{+})$. For example, in your Algorithm 1 or 2, we can replace $p(s+)$ with arbitrary distribution (even the distribution we create), then the whole derivation still hold. I believe it is reasonable to try to distinguish $p(s+)$ with $p(s+|s_t, a_t)$, but did you compare with any other prior distribution of $p(s+)$?

[clarity and theoretical soundness]
My major concern of the paper is the clarity which I will explain in the sequel.
- Notation.
 The main notation of the future state density function is abused the notation of $p$. I think in Definition 1 the right hand side you are defining a new condition density function, not a new future state. So I would recommend to write it as:
$$p_{+}(s'|s,a) = (1-\gamma)\sum_{t=0}^{\infty}\gamma^{t}p_{t}(s'|s,a),$$
where $p_t(s'|s,a)$ is the distribution if $s'$ occur $t$ step after the occurrence of the $s,a$ pair.
Many notations in the main text has this kind of problem where it is hard to distinguish the future distribution with the original distribution.
- Main message in section 5 is not clear.
Algorithm box 1 is confusing. It seems to me that we have double terms of $s_{t,+}$, and they are all used in the loss function (F=1 and F=0). If I understand correctly, we should put $s_{t,+,0} \sim p(s_{t,+}$ (again I don't like this notation, p can be any other meaning) and $s_{t,+,1} \gets s_{t+\delta}$, and the loss function for $F=0$ uses $s_{t,+,0}$ and $F=1$ uses $s_{t,+,1}$.
And without looking at the paragraph in details, it is hard to tell what is $F$ at the first glance. I feel like you can rewrite this section a little bit by motivating the reader why we consider using a classifier.
-Minor.
  1. Remark 1 should add some assumption, otherwise we can always think tabular case is a special case of in the continuous state space, where the probability mass function is not 0.
  2. Eq.(1) should be $p(F=1|s_t, a_t, s_{t,+}) = $XXX, missing a $s_{t,+}$ in left hand side. Similar for the next line.
  3.The sentence after Eq.(2), why we can get rid of estimating the marginal density? Not clear and no explanation.
  4. Eq.(4) missing an expectation for $E_{s_{t+1}, a_{t+1}}$.
  5. Eq.(5), we should put $F(\theta, \pi) = $ in the equation not a line above.
  6. Eq.(5), the subscript of expectation is not standard (compared to Eq.(3)).
  7. Unclear reason why we want to stop gradient in Eq.(7) and Algorithm 3. Is that stabilized the learning process?

[related work]
In the first line of the page 2, I think "contrastive approach" is not accurate. I didn't see any contrastive objective in the paper.
I feel like the density based methods in off-policy evaluation(e.g. Liu et. al. 2018 and DICE family (e.g. Nachum et. al. 2019)) is more relevant to value based method, where we can set the reward function as the indicator function and estimate the average reward of the policy $\pi$. Their methods are also based on a recursive objective, I think a comparison between that will be beneficial to further clarify your contribution.

[reference]
1. Qiang Liu, Lihong Li, Ziyang Tang, Dengyong Zhou. Breaking the Curse of Horizon: Infinite-Horizon Off-Policy Estimation
2. Ofir Nachum, Yinlam Chow, Bo Dai, Lihong Li. DualDICE: Behavior-Agnostic Estimation of Discounted Stationary Distribution Corrections

---

> ### Author Response · Authors · 2020-11-14
> **Response to R5**
>
> Thank you for the detailed review, and for all the suggestions. We believe that the two main concerns are (1) clarity and (2) the relationship with off-policy evaluation, but please let us know if there are other major concerns. As we describe below, we have revised the paper (1) to address all the clarity issues noted and (2) to clarify that off-policy evaluation is solving a different problem (policy evaluation vs policy search). We believe that these changes and additions address all of the major issues. Please let us know if there are other modifications or improvements that are important for the paper.
>
> **Clarity**: We have revised the paper to take into account the suggested writing changes:
> * Updated notation to use $p_+(s_{t+} \mid s_t, a_t)$ for the future state distribution and $p_\Delta(s_{t+\Delta} \mid s_t, a_t)$ to refer to the distribution of states exactly $\Delta$ steps in the future.
> * Updated notation to use $s_{t+}^{(0)}$ and $s_{t+}^{(1)}$ to clarify how goals are sampled in Section 5 and Algorithm 1.
> * Added two sentences to the start of Section 5.1 to motivate why we learn a classifier.
> * Slightly reworded Remark 1 to emphasize the assumptions. Please let us know if we haven't addressed the concern here.
> * Added the missing $s_{t+}$ on the LHS of Eq 1 and in the following line.
> * Added a sentence after Eq 2 explaining why we will be able to ignore the $p(s_{t+})$ term when learning a policy.
> * Added the missing expectation on the RHS of Eq 4
> * Moved $F(\theta, \pi)$ into Eq. 5
> * We didn't update the subscript in Eq 5 to match Eq 3 because we think that (1) unlike Eq. 3 there's no ambiguity which variable is sampled from which distribution and (2) adding additional math to the subscript might make it harder to read. Let us know if you have other ideas here.
> * Added a sentence explaining the stop_gradient after Eq. 7. The stop gradient is just a reminder that, when taking the gradient of an importance-weighted estimator, one should not take gradients through the importance weights.
> * We revised the related work to not refer to our method as a "contrastive approach."
>
> We welcome additional suggestions for improving clarity.
>
> **Off-Policy Evaluation**: We have also updated the paper to include a discussion of off-policy evaluation [Liu 18, Nachum 29]. Our method is similar to these prior papers in that we predict the future state distribution in an off-policy manner. Unlike these prior papers, our method learns a policy to control the future state distribution, leading to a goal-conditioned RL algorithm. These prior methods do not describe how to learn a policy. While these prior papers focus on using the predicted future state distribution to estimate the expected reward, our method does not require a hand-designed reward function at all.
>
> > "but did you compare with any other prior distribution of $p(s_{t+})$?"
>
> Our experiments used a few different choices of the prior distribution $p(s_{t+})$. For the goal-conditioned RL experiments in Fig. 3, $p(s_{t+})$ was a uniform distribution over previously-visited states (those stored in the replay buffer). For the prediction experiments in Fig. 2, $p(s_{t+})$ was a uniform distribution over the states visited by the expert policy.

---

> > ### Author Response · Authors · 2020-11-20
> > **Have the clarifications and revisions addressed the reviewer's concerns?**
> >
> > With the rebuttal deadline just a few days away, we wanted to follow up with the reviewer to confirm that the clarifications and paper revisions discussed above have addressed all the concerns raised in the review. We kindly ask the reviewer to let us know if they have any outstanding questions or concerns so that we can address these.

---

### Decision · Program_Chairs · 2021-01-07
**Final Decision**

**Decision:**

Accept (Poster)

**Comment:**

**Overview**: This paper provides a new clustering-based method to predict future probability density of a policy. It provides comparable performance to prior Q-learning-based methods, but without careful hyper-parameter tuning.

**Pro**: The method of using clustering to estimate future density is novel. Both theory and experiments appear solid. In the rebuttal phase, the authors convinced all the reviewers by addressing their concerns. The reviewers unanimous tend to acceptance.

**Con**: The reviewers had many concerns before the rebuttal. But these were addressed by the authors.

**Recommendation**: The C-learning method proposed in this paper is novel and can be potentially useful in practice. Both theory and experiments are solid and convincing. Hence the recommendation is accept.